# POLAR: AUTOMATING CYBER THREAT PRIORITIZATION THROUGH LLM-POWERED ASSESSMENT

## ABSTRACT

The rapid expansion of the cyber threat landscape, with over 11,000 new vulnerabilities reported in 2024 alone, has intensified the need for effective threat prioritization. Existing approaches, from rule-based systems to machine learning models, struggle with scalability, distribution shift, and context-independent scoring, often mis-ranking threats in dynamic exploitation environments. In this work, we present POLAR, an LLM-based framework that automates cyber threat prioritization across four sequential stages: Triage, Static Analysis, Exploitation Analysis, and Mitigation Recommendation. POLAR leverages LLM reasoning to transform unstructured threat intelligence into structured severity metrics, forecast exploitation likelihood using temporal narratives, and generate prioritized mitigation strategies. Through extensive evaluations, we highlight that POLAR not only improves prioritization accuracy for various cyber threats in the wild but also provides instructive outputs that assist analyst decision-making, which bridges the gap between automated threat hunting and real-world security practices.

## 1 INTRODUCTION

The cyber threat landscape has expanded dramatically as evidenced by over 11,000 newly reported vulnerabilities in 2024 (a 38% rise) compared to 2023 (NIST NVD, 2019). Amid rising threats, security analysts must perform **Prioritization** to assess which threats require immediate attention and resources to be patched or mitigated. In practice, prioritization involves ranking concurrent threats by urgency, potential impact, and possibility of exploitation. Guidelines defined by Common Vulnerability Scoring System (CVSS) (FIRST, 2019a;b; NIST NVD, 2019) and the Exploit Prediction Scoring System (EPSS) (Jacobs et al., 2021b; 2019) provide quantitative measures (not integrated tools) of exploitability, exposure, asset value, and attacker capability, facilitating organizations to follow standardized rubrics or guidelines for their prioritization developments.

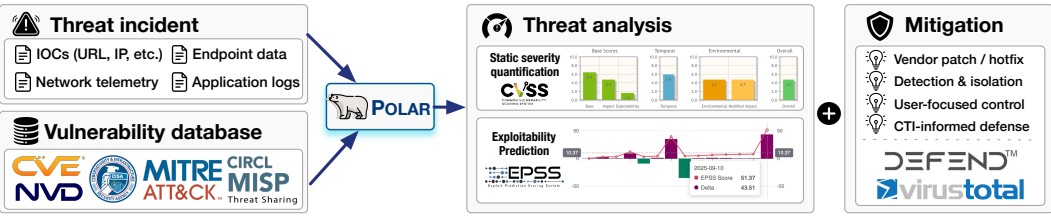

**Figure 1:** Overview of POLAR as an automated threat prioritization framework: it integrates real-world threat incidents with external databases to assess threats and recommend mitigations.

Existing prioritization tools remain constrained in addressing the scale and complexity of the modern threat landscape. Rule-based systems such as Snort (Roesch et al., 1999), Suricata (White et al., 2013), or SIEM/SOAR platforms (Sancho et al., 2020) rely on static signatures and predefined heuristics, making them brittle in ranking evolving threats such as polymorphic malware or living-off-the-land techniques (Barr-Smith et al., 2021; Dong et al., 2023; Liu et al., 2024), where the lack of predefined patterns leads to misaligned prioritization. AI approaches, including classifiers trained on common vulnerabilities and exposures (CVEs) (Vulnerabilities, 2005; Zhan et al., 2021) or anomaly detectors applied to logs (Chen et al., 2021; Zhang et al., 2022), provide better flexibility but suffer from

distribution shift on novel cyber threat intelligence (CTI) sources (e.g., threat reports, dark-web chatter) (Ren et al., 2019; Hendrycks & Gimpel, 2016), thereby hindering effective prioritization that can keep pace with evolving exploitation efforts. Moreover, both traditional and ML-based systems often assume context independence, wherein prioritization scores are assigned to features such as CVSS vectors, alerts, or log signatures without considering their interdependencies within broader exploitation contexts (Ou et al., 2005; Sheyner et al., 2002). As a result, these systems frequently mis-rank zero-day vulnerabilities or require substantial data augmentation to maintain prioritization accuracy in rapidly shifting environments.

Recent advances in large language models (LLMs) offer a potential solution in cybersecurity activities. Unlike rule-based systems or ML classifiers, LLMs demonstrate strong flexibility on reasoning and generalization across heterogeneous environments (Deng et al., 2024). In domain-specific scenarios, LLMs further exhibit adaptability by ingesting contextual knowledge, thereby enabling security-centric analyses such as penetration testing (Deng et al., 2024) or attack path discovery (Prapty et al., 2024; Ou et al., 2005). More importantly, LLMs are capable of integrating signals into a broader narrative context, linking seemingly independent indicators (e.g., anomalous DNS traffic, suspicious user logins, and privilege-escalation attempts) into coherent attack chains (Uetz et al., 2024; Zhang et al., 2022). These advantages position LLMs as a strong candidate for automating the threat prioritization in the face of the ever-evolving threat landscape.

However, the general-purpose utility of LLMs is often criticized for their insufficiency as domain specialists (Ren et al., 2019; Fort et al., 2021). The lack of targeted customization opens a gap between threat prioritization and the current adoption of LLMs. We thus ask: **How can LLMs be customized to automate cyber threat prioritization?**

In this work, we address this question through the following contributions:

**Approach.** We design and implement POLAR (Figure 1), a LLM-based framework that automates cyber threat prioritization from incident analysis to threat assessments and mitigation recommendation. POLAR leverages LLM reasoning in each stage to process unstructured threat knowledge (e.g., advisories, logs, KEV entries), quantify severity through CVSS metrics, forecast exploitation likelihood via temporal narratives, and finally, generate actionable mitigation strategies. This end-to-end pipeline fills a gap in the automation of cyber defense operations.

**Evaluation.** We conduct experiments of POLAR using real-world vulnerability databases enriched with temporal threat events. Our evaluation covers a broad range of CVEs and incorporates cross-referenced evidence from Exploit-DB (Offensive Security, 2025), CISA KEV (Cybersecurity and Infrastructure Security Agency (CISA), 2024), VirusTotal (VirusTotal, 2004), and vendor advisories, enabling benchmarking against industry-leading LLMs.

**Findings.** Our findings reveal important insights: (1) POLAR consistently improves prioritization tasks, showing the necessity of specialized automation in cybersecurity contexts. (2) POLAR presents stable performance even when processing large batches of entangled or heterogeneous incidents. (3) POLAR demonstrates improvement in high-severity threats that are undergoing intensive exploitation, with more robust and timely prioritization than baseline models and human analysis. These improvements highlight POLAR's ability to adapt to rapidly evolving threat dynamics and reduce the likelihood of mis-prioritizing critical vulnerabilities. To support future research, we have released our code at: `https://anonymous.4open.science/r/LLM-Prioritization-01D2/`.

## 2 MOTIVATING EXAMPLE

> **An Incident**. A vulnerability in the widely used `Jenkins` automation server allowed *unauthenticated attackers* to read arbitrary files from the server's filesystem via crafted *CLI* requests. Within days, proof-of-concept (PoC) exploits appeared on *Exploit-DB* and *GitHub*, and attacker campaigns weaponized the flaw for *initial access* and *data exfiltration* in DevOps pipelines. The challenge was compounded by customized `Jenkins` plug-ins and nonstandard configurations, making exploitation risk vary across deployments. Security teams therefore had to quickly assess imminent risks of this vulnerability and prioritize mitigation amid hundreds of concurrent alerts.

**Human Team Prioritization.** Relying solely on manual prioritization is highly inefficient. Analysts would need to manually parse long advisories, vendor bulletins, and CTI reports, cross-reference

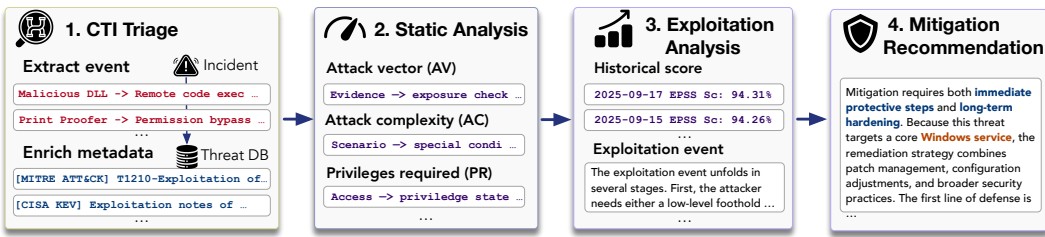

**Figure 2:** A running example of POLAR pipeline.

system configurations, and assess if the vulnerability's exploitation path was feasible in their environment. Given the rapid spread of PoCs and the adversaries' speed in adopting them, a manual workflow often results in missing the narrow window for preemptive patching. Furthermore, analysts face cognitive overload when correlating temporal signals (e.g., exploit-kit integration, chatter on criminal forums), making it extremely difficult to assign a timely exploitation probability.

**Conventional AI-based Prioritization.** Conventional AI models, such as classifiers or entity recognizers rely heavily on structured features (e.g., CVSS vectors, vulnerability age, vendor) and cannot capture emerging, context-specific signals like the release quality of a PoC or adversary targeting trends. These models are further constrained by incomplete metadata: if enrichment sources fail to provide patch status or exploit references, the prediction quality degrades significantly.

**LLM-powered Automation.** In contrast, POLAR integrates raw CTI feeds, advisories, and even unstructured forum posts to generate a structured temporal narrative of exploitation events. It reasons explicitly over inter-event gaps (e.g., disclosure → PoC release → KEV listing (Cybersecurity and Infrastructure Security Agency (CISA), 2024) → in-the-wild observation), evaluates weaponization quality from PoC descriptions, and judges adversary intent from chatter, all within a single unified reasoning loop. Finally, instead of producing a black-box score, the LLM outputs both a calibrated probability of exploitation in the next 30 days and a ranked set of mitigation actions tailored to the specific environment. This automation enables rapid, adaptive prioritization that human analysts or conventional models cannot feasibly achieve at the same scale and speed.

## 3 POLAR FRAMEWORK

**Overview.** We design POLAR to automate threat prioritization through four sequential stages, as illustrated in Figure 2: (1) In the **CTI Triage** stage, POLAR categorizes threat indicators and enriches the contexts with recorded metadata from vulnerability databases. (2) In the **Static Analysis** stage, threat contexts are mapped to CVSS metrics to provide a standardized baseline assessment of severity. (3) In the **Exploitation Analysis** stage, static scores are augmented with temporal evidence from CTI feeds, which capture historical exploitation activities and assist in forecasting future exploitation likelihood. (4) Finally, in the **Mitigation Recommendation** stage, POLAR fuses static and dynamic assessments into a composite report and links prioritized threats to actionable mitigation strategies.

### 3.1 CTI TRIAGE

CTI streams often arrive in unstructured formats (e.g., logs, advisories, reports) and may involve multiple overlapping events. POLAR performs CTI Triage to preprocess raw incidents into structured and separable threat instances. We consider two common scenarios: (1) a set of incidents **originating from distinct events or affecting different systems concurrently**, which can be assessed in parallel; and (2) a collection of threats **intertwined within the same event**, which must first be disentangled into individual components before prioritization.

Formally, let $\mathcal{I} = \{i_1, i_2, \ldots, i_N\}$ denote the set of raw incidents ingested at time $t$. Each incident $i_j$ is associated with a feature bundle $\mathbf{x}_j$ consisting of observable threat indicators (e.g., CVE identifiers, IP/domain artifacts, malware signatures) and contextual attributes (e.g., timestamps, affected assets, source feed). The triage step applies a mapping function $\phi : \mathcal{I} \mapsto \mathcal{T} = \{\tau_1, \tau_2, \ldots, \tau_M\}$ where each $\tau_k$ represents a structured *threat instance* ready for subsequent analyses. The mapping $\phi$ consists of two phases:

**Event Separation.** When multiple threats co-occur in the same raw incident $i_j$, POLAR directly prompts LLMs to disentangle $i_j$ into a set of indicators (i.e., metadata) $\{\mathbf{x}_{j_1}, \mathbf{x}_{j_2}, \ldots, \mathbf{x}_{j_L}\}$. Each sub-indicator is contextualized by natural language reasoning to be separated to different threats (e.g., distinguishing two CVEs reported in the same advisory but affecting different products). The resulting mapping function $\phi_{\text{LLM}}(\mathbf{x}_j) \mapsto \{\tau_1, \tau_2, \ldots, \tau_L\}$ outputs a set of structured threat instances that leverages LLMs' reasoning flexibility. We present the involved prompts in Appendix A.1.1.

**Metadata Enrichment.** Each extracted threat instance $\tau_k$ is contextualized by querying authoritative vulnerability databases such as NVD (National Institute of Standards and Technology (NIST), 2024) and MITRE ATT&CK (Strom et al., 2018). Let $\mathcal{D}$ denote the external knowledge sources. A retrieval operator $\psi : \tau_k \times \mathcal{D} \mapsto \mathbf{m}_k$ returns standardized metadata $\mathbf{m}_k$ (e.g., CVSS vector string, disclosure date, affected vendor/product). The enriched representation of a threat instance is thus defined as $\tilde{\tau}_k = (\tau_k, \mathbf{m}_k)$, which combines raw indicators with authoritative contextual knowledge. Appendix A.1.2 details the considered metadata and sourced databases.

---

**Example 1: CTI Triage with Metadata Enrichment (entangled example in Appendix A.1.3)**

**Raw Incident** ($i_k \in \mathcal{I}$)**:** In Windows 11, the print spooler service can be abused by an authenticated remote attacker to load a DLL through a crafted DCERPC request, resulting in remote code execution as NT AUTHORITY. This module uses the MS-RPRN vector which requires ...

**Threat Instance** ($\tau_k \in \mathcal{T}$)**:** {**Vendor**: `Microsoft Windows 11`, **Affected Components**: `Print Spooler`, **Campaign**: `Unauthorized permission bypass`, **Impact**: `Remote code execution`, **Attack Patterns**: `load malicious DLL via UNC/SMB`, ...}

**Enriched Metadata** ($\mathbf{m}_k$)**:** (1) Mapped MITRE ATT&CK: T1210 – Exploitation of Remote Services; T1068 – Exploitation for Privilege Escalation ... (2) From CISA KEV Catalog: Actively Exploited in the Wild, Exploitation Notes: ... (3) Mapped CVE: CVE-2021-34527 ...

---

**Output.** After triage, POLAR produces a structured set of enriched threat instances $\tilde{\mathcal{T}} = \{\tilde{\tau}_1, \tilde{\tau}_2, \ldots, \tilde{\tau}_M\}$ serving as inputs for the subsequent analyses.

## 3.2 STATIC ANALYSIS

Given triaged threat instances $\tilde{\mathcal{T}} = \{\tilde{\tau}_1, \tilde{\tau}_2, \ldots, \tilde{\tau}_M\}$, POLAR performs static analysis to quantify the severity of each threat $\tilde{\tau}_k \in \tilde{\mathcal{T}}$. The goal is to map triaged contexts into standardized *CVSS* metrics, which provide a standardized baseline for prioritization. Formally, for each threat instance $\tilde{\tau}_k = (\tau_k, \mathbf{m}_k)$, the static analysis computes a CVSS vector $\mathbf{v}_k \in \mathbb{R}^d$, where $d$ is the number of base metrics in CVSS (e.g., Attack Vector, Attack Complexity, Privileges Required, User Interaction, Confidentiality Impact, Integrity Impact, Availability Impact). A scoring function $\mathcal{S}_{\text{CVSS}} : \tilde{\tau}_k \mapsto \mathbf{v}_k$ is applied to obtain the CVSS vector, wherein each severity score $\mathbf{v}_k^{(i)} \in [0, 10]$.

**Metric-Specific Workflows.** Notably, each CVSS metric $\mathbf{v}_k^{(i)}$ focuses on specific CTI evidence and is therefore determined through a dedicated workflow derived from the triaged instance $\tilde{\tau}_k$. Specifically: **Attack Vector (AV)** is derived by cross-referencing the affected system type in $\mathbf{m}_k$ with exposure surfaces (e.g., network-facing service $\Rightarrow$ AV=Network). **Attack Complexity (AC)** is estimated by evaluating whether successful exploitation depends on environmental conditions (e.g., race conditions, system configurations). **Privileges Required (PR)** is inferred from metadata on authentication requirements, such as local user vs. administrator privileges. **User Interaction (UI):** is obtained by checking against advisory text to detect whether user action (e.g., opening a malicious document) is needed. **Impact Metrics (C – Confidentiality, I – Integrity, A – Availability):** is determined by analyzing the vulnerability description and mapped against CVSS guidelines for confidentiality, integrity, and availability. Detailed analytical steps are provided in Appendix A.2.

Formally, for each CVSS metric $\mathbf{v}_k^{(i)}$, POLAR constructs an evidence set $\mathcal{E}_k^{(i)} = \{(s_\ell, u_\ell)\}$ where $s_\ell$ is a text span (from the triaged instance $\tau_k$ or metadata $\mathbf{m}_k$) and $u_\ell$ is its source (e.g., vendor advisory, CTI feed). We adopt an LLM-assisted classifier $f_{\text{LLM}}^{(i)}$ maps $\mathcal{E}_k^{(i)}$ to a metric class $v_k^{(i)}$ in the metric's domain $\mathcal{Y}^{(i)}$: $v_k^{(i)} = f_{\text{LLM}}^{(i)}(\mathcal{E}_k^{(i)}, D_{\text{CVSS}})$, where $v_k^{(i)} \in \mathcal{Y}^{(i)}$. with official instructions $D_{\text{CVSS}}$ from CVSS (FIRST, 2019a;b). An example of $f_{\text{LLM}}^{(i)}(\cdot)$ is:

---

**Example 2: Implementation of $f_{\text{LLM}}^{(i)}(\cdot)$ (additional examples in Appendix A.2)**

Deciding **Attack Vector** $\in$ {Network, Adjacent, Local, Physical} (same incident as **Example 1**) :

**Evidence construction** ($\mathcal{E}_k^{(i)}$)**:** LLM searches $\tilde{\tau}_k$ to localize text spans $s$: "`authenticated remote attacker`", "`crafted DCERPC request`" ... and sources $u$: `Vendor advisory`...

**Exposure check:** LLM verifies service listens over the network (RPC over SMB/DCERPC).

**Conflict resolution:** If both "`local`" and "`remote`" cues appear, LLM selects the highest exposure that is explicitly supported by protocol evidence (DCERPC over network).

**Output: Attack Vector** = Network given multiple high-trust remote protocol spans.

---

**Output.** The static analysis stage generates a standardized CVSS vector $\mathbf{v}_k$ for each triaged threat, following the CVSS format (e.g., `AV:N/AC:L/PR:L/UI:N/S:C/C:H/I:H/A:H`). We then apply the official CVSS scoring algorithm to compute the overall severity score. Relying on LLMs to directly estimate the score would only introduce additional error.

### 3.3 EXPLOITATION ANALYSIS

Static CVSS scores provide a standardized baseline for severity; however, they do not capture the dynamic nature of real-world exploitation. Threats with high CVSS scores may never be exploited (e.g., CVE-2010-20123), while those with moderate scores can also be rapidly weaponized once proof-of-concept (PoC) exploits or adversarial campaigns emerge (e.g., CVE-2024-57785). To address these dynamics, POLAR performs **Exploitation Analysis** by retrieving temporal evidence from CTI feeds (e.g., Exploit-DB, CISA KEV, VirusTotal, full list in Appendix A.3) and forecasting exploitation likelihood.

Formally, for threat incident $\tilde{\tau}_k \in \tilde{\mathcal{T}}$, let the retrieved temporal evidence set as: $\mathcal{E}_k^{\text{temp}} = \{(t_1, e_1), (t_2, e_2), \ldots, (t_{L_k}, e_{L_k})\}$, where each tuple $(t_i, e_i)$ represents an exploitation-related event $e_i$ (e.g., exploit publication, public PoC release, observed in-the-wild exploitation) at timestamp $t_i$. The goal is to produce a predictive exploitation score $p_k \in [0, 1]$ representing the probability that threat $\tilde{\tau}_k$ will be exploited in the next 30 days (a common metric for EPSS prediction system (Jacobs et al., 2021b). We also evaluate on the alternative setting in §4.3.

**Temporal Modeling.** To estimate $p_k$, POLAR uses the triplet $x_k := (\tilde{\tau}_k, \mathbf{v}_k, \mathcal{E}_k^{\text{temp}})$ and to reason over a *temporal narrative* of $\mathcal{E}_k^{\text{temp}}$ (events sorted by time with inter-event gaps, e.g., CVE publication $\rightarrow$ PoC release $\rightarrow$ KEV listing $\rightarrow$ in-the-wild reports), enriched with $\mathbf{v}_k$ and basic asset/context metadata from $\tilde{\tau}_k$. POLAR aims to sequentially (i) identify exploitation signals (e.g., public PoCs), (ii) estimate exposure and mitigation frictions (e.g., prevalence, default configurations, patch availability), and (iii) judge adversary interest (e.g., by vendor advisories) to determine the $p_k$.

**Output.** POLAR outputs a single scalar $p_k \in [0, 1]$ representing "probability of observed exploitation in the future (e.g., next 30 days)," which aligns with practical interests (Jacobs et al., 2021a) to predict probability of recent-period exploitation activity.

### 3.4 MITIGATION RECOMMENDATION

After static analysis and exploitation forecasting, each threat instance is integrated as $(\tilde{\tau}_k, s_k, p_k)$. POLAR then recommends actionable mitigation strategies by linking $\hat{\tau}_k$ with mitigation knowledge (e.g., patches, configuration workarounds, threat intelligence advisories, and organizational asset criticality) and by leveraging $(s_k, p_k)$ to determine mitigation priorities. Specifically:

**Mitigation Retrieval.** POLAR uses the threat identifier and metadata from $\hat{\tau}_k$ (i.e., CVE ID, affected product, and software version) to locate relevant mitigations $a_k$ through an LLM-based retrieval process: $a_k = f_{\text{LLM}}^{\text{ret}}(\hat{\tau}_k, \mathcal{D}_{\text{rem}})$ where $f_{\text{LLM}}^{\text{ret}}$ is implemented as a structured retrieval-and-reasoning prompt. The prompt (detailed in Appendix A.4) is designed to match $\tilde{\tau}_k$ against authoritative external knowledge bases $\mathcal{D}_{\text{rem}}$, including NVD advisories (patch releases), MITRE ATT&CK mitigations (technique-specific defenses), CISA KEV (known exploited vulnerabilities with mitigation notes), and vendor security bulletins (vendor-supplied patches, hotfixes, and configuration workarounds).

**Table 1:** Evaluation of CTI triage performance (F1 of generated and ground-truth incidents or metadata) across POLAR-enhanced cybersecurity agents and general-purpose LLM backbones. The reported numbers in each cell indicate the POLAR-enhanced result and the increase (↑) or decrease (↓) compared with the backbone model.

| Model | | Incident Extraction | | Metadata Enrichment | | | | |
|---|---|---|---|---|---|---|---|---|
| | | Single Threat | Entangled Threats | CVE ID | TTP | Exploit Status | Affected Sys | Disclosure |
| **Cybersecurity-specialized agent** | | | | | | | | |
| **POLAR +** | **FS** | 0.793 (↑ 0.091) | 0.656 (↑ 0.103) | 0.715 (↑ 0.112) | 0.679 (↑ 0.127) | 0.772 (↑ 0.089) | 0.921 (↑ 0.149) | 0.805 (↑ 0.082) |
| | **LY** | 0.786 (↑ 0.084) | 0.674 (↑ 0.121) | 0.697 (↑ 0.094) | 0.666 (↑ 0.114) | 0.766 (↑ 0.083) | 0.844 (↑ 0.042) | 0.794 (↑ 0.071) |
| | **ZY** | 0.811 (↑ 0.109) | 0.686 (↑ 0.133) | 0.724 (↑ 0.121) | 0.688 (↑ 0.136) | 0.781 (↑ 0.098) | 0.863 (↑ 0.061) | 0.810 (↑ 0.087) |
| **General-purpose LLM** | | | | | | | | |
| **POLAR +** | **GE2.5** | 0.895 (↑ 0.193) | 0.834 (↑ 0.281) | 0.827 (↑ 0.257) | 0.809 (↑ 0.242) | 0.925 (↑ 0.242) | 0.855 (↑ 0.023) | 0.814 (↑ 0.191) |
| | **O4M** | 0.883 (↑ 0.214) | 0.857 (↑ 0.304) | 0.864 (↑ 0.261) | 0.854 (↑ 0.302) | 0.946 (↑ 0.263) | 0.865 (↑ 0.163) | 0.822 (↑ 0.239) |
| | **G5** | 0.916 (↑ 0.251) | 0.894 (↑ 0.341) | 0.826 (↑ 0.173) | 0.864 (↑ 0.312) | 0.884 (↑ 0.303) | 0.783 (↑ 0.018) | 0.937 (↑ 0.274) |
| | **L70B** | 0.873 (↑ 0.171) | 0.796 (↑ 0.243) | 0.839 (↑ 0.076) | 0.817 (↑ 0.265) | 0.897 (↑ 0.214) | 0.805 (↑ 0.103) | 0.787 (↑ 0.401) |
| | **C4.1** | 0.825 (↑ 0.223) | 0.816 (↑ 0.313) | 0.835 (↑ 0.212) | 0.778 (↑ 0.326) | 0.767 (↑ 0.284) | 0.874 (↑ 0.142) | 0.824 (↑ 0.261) |
| | **Q3** | 0.903 (↑ 0.201) | 0.824 (↑ 0.271) | 0.857 (↑ 0.104) | 0.843 (↑ 0.291) | 0.924 (↑ 0.241) | 0.827 (↑ 0.121) | 0.835 (↑ 0.332) |

**Mitigation Prioritization.** POLAR leverages LLM reasoning to suggest a prioritization order using $(s_k, p_k)$ for each threat. POLAR compares threats in groups to produce an ordered ranking of recommended actions (detailed implementation in Appendix A.4). This approach avoids rigid weightings and allows adaptive prioritization based on both quantitative risk scores and qualitative mitigation constraints (e.g., ease of patching, business impact, compensating controls).

**Output.** This stage outputs a ranked list of actionable mitigation strategies for triaged threat instances, as described in §3.1.

## 4 EXPERIMENT

We perform extensive evaluations on POLAR, aiming to answer the following research questions: **RQ₁:** Can POLAR accurately triage threat indicators and enrich metadata from raw incidents? **RQ₂:** How effective is POLAR in conducting static analysis (i.e., scoring CVSS metrics)? **RQ₃:** Can POLAR reliably forecast exploitation likelihood? **RQ₄:** Does POLAR effectively retrieve and prioritize actionable mitigations?

**LLMs.** As baseline and backbone of POLAR, we evaluate with six representative LLMs: Gemini-2.5-Pro (**GE2.5**), GPT-o4-mini-high (**O4M**), GPT-5 (**G5**), LLaMA-3-70B-Instruct (**L70B**), Claude Opus 4.1 (**C4.1**), and Qwen3-30B-A3B-Instruct-2507 (**Q3**). We also evaluate three LLM-based cybersecurity-specialist agents: Foundation-Sec-8B-Instruct (**FS**) (Singer, 2025), Lily-Cybersecurity-7B (**LY**) (Labs, 2024), and ZySec-7B (**ZY**) (ZySec AI, 2024).

**Data.** We collect raw threat incidents from high-fidelity vendor advisories (e.g., Microsoft, Cisco, Oracle), security blogs and reports (e.g., Mandiant (Mandiant, 2004)), public CTI aggregators (e.g., KEV Catalog (Cybersecurity and Infrastructure Security Agency (CISA), 2024)), and exploit repositories (e.g., Exploit-DB (Offensive Security, 2025), GitHub PoCs). Each incident is mapped into structured evaluations across the four stages of POLAR: (i) Triage/Metadata is grounded against authoritative repositories such as CVE/NVD (CVE Program, 2024) and MITRE ATT&CK (Strom et al., 2018), (ii) CVSS scores are validated using official NVD-assigned vectors, (iii) exploitation probabilities are referenced from the FIRST.org EPSS dataset (Jacobs et al., 2021b), and (iv) Mitigation strategies are drawn from NVD advisories, KEV mitigation notes, and vendor security updates. We standardize raw incidents into stage-specific tasks (e.g., extracting metadata from advisories), and treat the official repository assignment as the reference answer in cases of ambiguity. Details of quality control and statistics are reported in Appendix B.1.

### 4.1 CTI TRIAGE PERFORMANCE (RQ₁)

**Setting.** The evaluation of triage performance covers two aspects: (1) whether LLMs can accurately extract threat instances from raw incidents (including both single and entangled threats), and (2) whether the enriched metadata align with references.

**POLAR improves backbone models at varying rates.** Our evaluations in Table 1 show consistent improvements across all models when equipped with POLAR. For cybersecurity-specialized agents, the gains are modest, as these agents already embed domain-specific heuristics and curated knowledge,

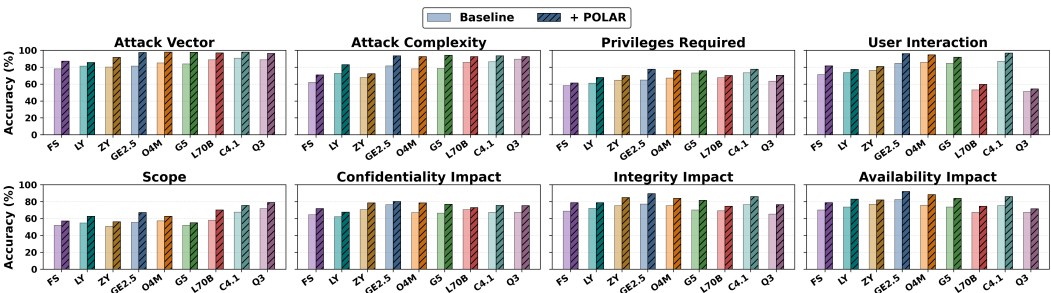

**Figure 3:** Static analysis performance (accuracy) on baselines and POLAR.

which reduce the marginal benefit of additional specialization. In contrast, general-purpose LLMs achieve substantially larger gains once integrated with POLAR. This is because general-purpose models lack direct grounding in authoritative CTI repositories. By imposing structured triage and metadata enrichment workflows, POLAR bridges these gaps, demonstrating that neither domain specialization alone nor unconstrained reasoning is sufficient.

**Triage improvement is influenced by the integration of security knowledge.** The most substantial gains appear in tasks that demand knowledge-grounded reasoning, such as separating entangled advisories and aligning threat instances with ATT&CK techniques or exploitation status. These outcomes highlight POLAR's strength in transforming crowdsourced CTI into well-organized, actionable intelligence. Moreover, the improvements hold consistently across heterogeneous LLM backbones, showcasing POLAR as a transferable automation framework for CTI operations. This finding reinforces a critical design principle: **effective prioritization requires frameworks that balances the flexibility of LLM reasoning with domain-grounded workflow.**

Due to space constraints, we defer the in-depth analysis of POLAR's gains and gaps to B.2.1.

## 4.2 STATIC ANALYSIS EFFECTIVENESS (RQ$_2$)

**Setting.** Static analysis covers both *classification* of CVSS metrics and *scoring* of overall severity. We evaluate per-metric classification accuracy and RMSE against official NVD base scores.

**POLAR's consistency.** As shown in Figure 3 and 4, POLAR improves accuracy across all eight CVSS metrics and reduces RMSE consistently. Gains are most obvious in context-heavy metrics such as Attack Vector and Attack Complexity (∼20% improvements on some backbones like Gemini), while even straightforward metrics such as Scope and Impacts show steady lifts. These results confirm POLAR's ability to deliver more reliable static severity assessments.

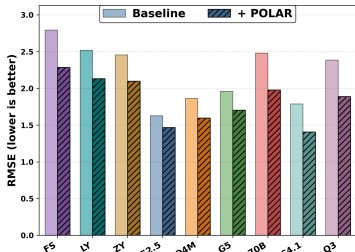

**Figure 4:** Overall scoring performance (RMSE) in static analysis.

**POLAR's integrative ability.** POLAR enhances static analysis by combining retrieval-grounded evidence (from CTI triage) with structured reasoning, which mitigates common misclassifications (e.g., distinguishing local vs. network attack vectors) and enables analysts to trust CVSS classifications for prioritization. POLAR also lowers the risk of overlooking critical vulnerabilities or misallocating patching resources. In practice, this translates to more reliable alignment between threat scoring and real-world risks. We provide more details in B.2.2.

## 4.3 EXPLOITATION ANALYSIS RELIABILITY (RQ$_3$)

**Setting.** We evaluate POLAR under: (1) Different exploitation trends (stable, steadily increasing/decreasing, and abrupt changes triggered by events such as proof-of-concept release), with results in Table 2. (2) Different study lengths of exploitation history, with results in Figure 5.

**POLAR's robustness in variant forecasting scenarios.** For exploitation trends shown in Table 2, POLAR consistently improves prediction across all trend types. While baseline models can track

**Table 2:** Exploitation forecasting by trend type (using a 1-year exploitation history to predict the next 30/90 days). Each cell reports POLAR-enhanced results along with the change relative to baseline models: ↓ indicates reduced RMSE, ↑ indicates improved "DirAcc" (accuracy of direction prediction), which measures whether the trend is increasing, decreasing, or stable. Results for cybersecurity-specialized agents are provided in Table 5.

| Model | | 30 Days | | 90 Days | |
|---|---|---|---|---|---|
| | | RMSE ($\times 10^{-3}$) ↓ | DirAcc ↑ | RMSE ($\times 10^{-3}$) ↓ | DirAcc ↑ |
| Monotonic trend of exploitation (increasing/decreasing) | | | | | |
| POLAR + | GE2.5 | 1.183 (↓ 0.543) | 0.889 (↑ 0.105) | 1.479 (↓ 0.444) | 0.813 (↑ 0.078) |
| | O4M | 1.275 (↓ 0.454) | 0.867 (↑ 0.089) | 1.596 (↓ 0.372) | 0.795 (↑ 0.065) |
| | G5 | 1.389 (↓ 0.341) | 0.852 (↑ 0.078) | 1.738 (↓ 0.279) | 0.781 (↑ 0.057) |
| | L70B | 1.622 (↓ 0.108) | 0.814 (↑ 0.061) | 2.029 (↓ 0.089) | 0.747 (↑ 0.045) |
| | C4.1 | 1.098 (↓ 0.629) | 0.904 (↑ 0.139) | 1.373 (↓ 0.515) | 0.829 (↑ 0.102) |
| | Q3 | 1.532 (↓ 0.198) | 0.826 (↑ 0.064) | 1.916 (↓ 0.162) | 0.759 (↑ 0.047) |
| Stable trend of exploitation (no significant increasing/decreasing trend) | | | | | |
| POLAR + | GE2.5 | 0.172 (↓ 0.093) | 0.482 (↑ 0.084) | 0.216 (↓ 0.076) | 0.418 (↑ 0.065) |
| | O4M | 0.198 (↓ 0.067) | 0.451 (↑ 0.061) | 0.249 (↓ 0.055) | 0.391 (↑ 0.047) |
| | G5 | 0.215 (↓ 0.051) | 0.438 (↑ 0.059) | 0.269 (↓ 0.042) | 0.379 (↑ 0.046) |
| | L70B | 0.247 (↓ 0.019) | 0.417 (↑ 0.040) | 0.309 (↓ 0.015) | 0.361 (↑ 0.031) |
| | C4.1 | 0.163 (↓ 0.103) | 0.492 (↑ 0.094) | 0.204 (↓ 0.084) | 0.426 (↑ 0.073) |
| | Q3 | 0.231 (↓ 0.035) | 0.427 (↑ 0.034) | 0.290 (↓ 0.028) | 0.369 (↑ 0.026) |
| Abrupt change of exploitation | | | | | |
| POLAR + | GE2.5 | 63.582 (↓ 38.750) | 0.603 (↑ 0.131) | 78.944 (↓ 31.735) | 0.524 (↑ 0.103) |
| | O4M | 69.885 (↓ 32.448) | 0.573 (↑ 0.100) | 86.722 (↓ 26.588) | 0.497 (↑ 0.078) |
| | G5 | 75.212 (↓ 27.121) | 0.546 (↑ 0.074) | 93.389 (↓ 22.219) | 0.474 (↑ 0.058) |
| | L70B | 84.442 (↓ 17.891) | 0.489 (↑ 0.017) | 104.828 (↓ 14.653) | 0.424 (↑ 0.013) |
| | C4.1 | 59.540 (↓ 42.792) | 0.625 (↑ 0.152) | 73.920 (↓ 35.068) | 0.542 (↑ 0.118) |
| | Q3 | 82.821 (↓ 19.512) | 0.496 (↑ 0.025) | 102.841 (↓ 15.981) | 0.430 (↑ 0.019) |

gradual increases or decreases, they often fail to respond to sudden spikes or drops in exploitation activity. In contrast, POLAR captures abrupt changes by linking temporal signals to the broader threat narrative, making it more reliable for predicting sharp adjustments in probability estimates.

**Exploitation contexts may not always be helpful.** For historical exploitation, Figure 5 shows that using excessively long windows does not necessarily improve forecasting of future exploitation probabilities. Earlier records can introduce noise from outdated or irrelevant events (e.g., prior exploitation campaigns targeting unrelated product versions), thereby diluting the signal of current exploitation activity. Practically, this implies that **threat assessment should calibrate context horizons to operational timeframes (e.g., quarterly or semi-annual review cycles) rather than defaulting to maximal history.** Despite this influence, POLAR still demonstrates consistent improvements over baselines, thus highlighting its robustness even when contextual noise is present. We provide more discussions in Appendix B.2.3.

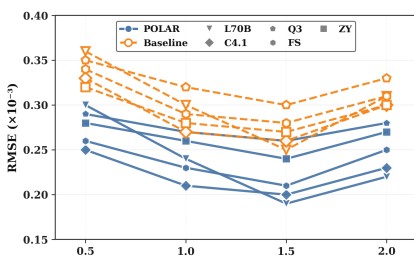

**Figure 5:** RMSE on varying study contexts (years of exploitation history).

### 4.4 MITIGATION RECOMMENDATION EFFICACY (RQ₄)

**Setting.** We evaluate mitigation efficacy by (1) the correctness of prioritization order across threats using ranking-based metrics (NDCG@k, Kendall's $\tau$) and (2) the F1 of retrieved mitigation actions, focusing on four categories: CVE-to-patch mapping, ATT&CK mitigation alignment, mitigation notes (e.g., KEV notes), and vendor-specific advisories.

**POLAR's effectiveness in prioritizing mitigation.** Results in Table 3 show that POLAR generates more accurate mitigation prioritization than all baselines, especially under scenarios where multiple medium-severity threats compete for limited resources. Through reasoning over both severity $(s_k)$ and exploitation likelihood $(p_k)$, POLAR adapts prioritization beyond rigid scoring rules, aligning closely with ground-truth mitigation urgency. For retrieval, POLAR achieves higher performance across all four categories, with relevantly larger gains in ATT&CK mapping, which belongs to tasks that require contextual reasoning beyond keyword matching. Together, these findings highlight that POLAR not only retrieves more relevant mitigation knowledge but also transforms it into actionable and correctly ordered recommendations, thus reducing the wasted effort on low-risk threats while ensuring the high-risk ones are patched first. We provide more analyses in Appendix B.3.

**Table 3:** Evaluation of recommendation performance (F1 between generated and reference mitigation items from sourced databases) across POLAR-enhanced cybersecurity agents and general-purpose LLM backbones. The reported numbers in each cell indicate the POLAR-enhanced result and the increase (↑) or decrease (↓) compared with the backbone model.

| Model | | Prioritization Correctness | | Mitigation Retrieval | | | |
|---|---|---|---|---|---|---|---|
| | | NDCG@5 | Kendall's $\tau$ | CVE-to-Patch | ATT&CK Mapping | Mitigation Note | Vendor Advisory |
| **Cybersecurity-specialized agent** | | | | | | | |
| **POLAR +** | **FS** | 0.685 (↑ 0.083) | 0.553 (↑ 0.101) | 0.734 (↑ 0.113) | 0.635 (↑ 0.132) | 0.693 (↑ 0.091) | 0.625 (↑ 0.073) |
| | **LY** | 0.694 (↑ 0.092) | 0.564 (↑ 0.112) | 0.717 (↑ 0.096) | 0.624 (↑ 0.121) | 0.683 (↑ 0.081) | 0.616 (↑ 0.064) |
| | **ZY** | 0.705 (↑ 0.103) | 0.575 (↑ 0.123) | 0.742 (↑ 0.121) | 0.646 (↑ 0.143) | 0.704 (↑ 0.102) | 0.634 (↑ 0.082) |
| **General-purpose LLM** | | | | | | | |
| **POLAR +** | **GE2.5** | 0.793 (↑ 0.191) | 0.693 (↑ 0.241) | 0.844 (↑ 0.223) | 0.757 (↑ 0.254) | 0.845 (↑ 0.243) | 0.743 (↑ 0.191) |
| | **O4M** | 0.825 (↑ 0.223) | 0.735 (↑ 0.283) | 0.885 (↑ 0.264) | 0.796 (↑ 0.293) | 0.864 (↑ 0.262) | 0.784 (↑ 0.232) |
| | **G5** | 0.873 (↑ 0.271) | 0.783 (↑ 0.331) | 0.943 (↑ 0.322) | 0.815 (↑ 0.312) | 0.903 (↑ 0.301) | 0.823 (↑ 0.271) |
| | **L70B** | 0.765 (↑ 0.163) | 0.673 (↑ 0.221) | 0.824 (↑ 0.203) | 0.744 (↑ 0.241) | 0.814 (↑ 0.212) | 0.735 (↑ 0.183) |
| | **C4.1** | 0.834 (↑ 0.232) | 0.743 (↑ 0.291) | 0.904 (↑ 0.283) | 0.804 (↑ 0.301) | 0.884 (↑ 0.282) | 0.805 (↑ 0.253) |
| | **Q3** | 0.806 (↑ 0.204) | 0.714 (↑ 0.262) | 0.872 (↑ 0.251) | 0.774 (↑ 0.271) | 0.863 (↑ 0.261) | 0.785 (↑ 0.233) |

## 5 RELATED WORK

**Threat prioritization.** Traditional approaches to threat prioritization primarily relied on heuristic-driven models. Statistical and probabilistic methods, including Bayesian models and attack graphs, have been used to assess risk propagation and exploit likelihood within specific infrastructures (Ingols et al., 2009; Poolsappasit et al., 2012). Other approaches integrate expert knowledge with structured taxonomies, such as MITRE's ATT&CK framework, to map observed adversary techniques to potential system impacts (Strom et al., 2018). However, these methods generally emphasize static attributes and predefined rules, making them less effective in rapidly evolving threat environments.

**LLMs for cybersecurity tasks.** Surveys report broad LLM capability—alongside fragility—in code analysis, vuln detection/repair, and broader security workflows (Hasanov et al., 2024; Das et al., 2025; Zhou et al., 2025; Chen et al., 2025). Task-specific systems show gains in penetration testing via tool use and chain-of-thought orchestration (PentestGPT) (Deng et al., 2024), and in attack modeling via retrieval-augmented generation over threat knowledge bases (Prapty et al., 2024). Evaluation and deployment studies surface challenges around robustness, hallucination, and operational risk (McGraw et al., 2024; Keppler et al., 2024). Most prior efforts, however, target isolated tasks rather than end-to-end prioritization.

**LLM agents and automation.** LLM-as-agent paradigms combine reasoning, tools, and environment interaction for complex goals (Dong et al., 2023). In security, automated attack-graph construction and reasoning trace feasible paths and preconditions (Ou et al., 2005; Sheyner et al., 2002), while multi-agent designs specialize roles across assessment, exploitation prediction, and mitigation planning (Sancho et al., 2020). Retrieval-augmented agents and deep log-analysis models enhance detection under real-world data heterogeneity (Du et al., 2017; Zhang et al., 2022; Uetz et al., 2024). Unlike monolithic prompts or single-purpose tools, modular agents maintain global context while executing task-specific reasoning. POLAR follows this design: a multi-stage, retrieval-aware pipeline that integrates CVSS interpretability with evidence-conditioned temporal dynamics to support end-to-end threat prioritization.

## 6 CONCLUSION

We present POLAR, an LLM-based framework for automated cyber threat prioritization that integrates incident triage, severity scoring, exploitation forecasting, and mitigation recommendation into a unified pipeline. Through large-scale evaluations, we demonstrated that POLAR outperforms both general-purpose LLMs and specialized agents, particularly in handling dynamic exploitation trends and high-severity threats. By aligning closely with analyst reasoning while scaling to the volume and complexity of modern CTI, POLAR offers a practical solution for next-generation cyber defense automation.

ETHICS STATEMENT

This work does not raise ethical concerns. All experiments were conducted on publicly available cyber threat intelligence resources, standardized vulnerability databases (e.g., CVE, NVD, CISA KEV), and vendor advisories. No private, personal, or sensitive data were accessed during the research. The methodology and evaluation strictly comply with the respective licensing and usage policies of the data sources.

REPRODUCIBILITY STATEMENT

To support reproducibility, we have released the complete implementation of our framework, together with evaluation scripts and experimental configurations, through an anonymous GitHub repository detailed in Introduction. This repository includes detailed instructions for independent researchers to reproduce our results and extend our experiments.

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

## A    COMPLEMENTARY DETAIL OF POLAR

We provide additional details for POLAR, complementing §3.

### A.1    CTI TRIAGE

Complementing §3.1, we provide prompt design about how to partition entangled incident for downstream reasoning.

### A.1.1    DETAILS OF DISENTANGLEMENT

---

**Prompt: Threat Instance Disentanglement**

**System Instruction** You are a cybersecurity analyst LLM specialized in processing cyber threat intelligence (CTI) reports. Your task is to **disentangle entangled threat events** where multiple threats are described together. You must carefully extract, separate, and structure each distinct threat indicator, ensuring that no relevant information is lost. Follow these rules:

1. **Identify threat indicators:** Look for CVE identifiers, malware family names, IPs/domains, exploit techniques, affected products, and timestamps.
2. **Disentangle overlapping events:** If multiple threats are mentioned in one text, split them into **independent threat instances**. Each instance should only describe one main vulnerability, malware, or attack campaign.
3. **Preserve context:** Include any associated attributes (e.g., vendor, affected systems, attack vector, source reference).
4. **Be explicit and structured:** Output results in a structured JSON format with one entry per disentangled threat instance.
5. **Do not invent information:** Only use details explicitly present in the input text.
   . . . . . .

---

> **User Instruction** The following text may describe multiple threats entangled together. Please disentangle them into distinct structured threat indicators including · · ·
>
> **Input** {RAW_INCIDENT_TEXT}
>
> **Output** · · · · · ·

**Feasibility of LLM-automated disentanglement.** The disentanglement of threat indicators from a single raw incident is achievable by explicitly instructing LLMs through carefully designed prompts. As shown in our disentanglement prompt template, the system first constrains the model with rules: it must identify canonical threat indicators (such as CVEs, malware families, or infrastructure artifacts), separate entangled descriptions into independent instances, and preserve contextual metadata. By defining strict output placeholders, such as JSON objects for each instance, we mitigate the risk of free-form responses and enforce structured extractions suitable for downstream modules. This structured prompting approach leverages the LLM's semantic reasoning capability to parse natural-language advisories that often contain overlapping vulnerabilities or campaigns.

### A.1.2 Vulnerability Database of Platform for Metadata Enrichment

For each disentangled threat instance $\tau_k$, it is insufficient to rely solely on the raw indicators extracted from unstructured CTI text. To enable standardized analysis, we gather authoritative metadata from external vulnerability databases, government-maintained catalogs, and trusted threat intelligence platforms. Below we outline the primary categories of metadata used in our enrichment operator $\psi$, together with their relevance to downstream prioritization.

**CVE and Vulnerability Identification.** The cornerstone of enrichment is mapping each threat instance to a **unique identifier**, typically a Common Vulnerabilities and Exposures (CVE) ID (CVE Program, 2024). A CVE provides a canonical handle for correlating incident reports across vendors and feeds. Beyond the CVE label, metadata should include the vulnerability's publication date, last modified date, and record status (e.g., "RESERVED", "DISPUTED", or "REJECTED"). These attributes establish the maturity and stability of the reported issue.

**Tactics, Techniques, and Procedures (TTPs).** To align each vulnerability with attacker behavior, we query mappings to **MITRE ATT&CK TTPs** (Strom et al., 2018). Relevant fields include (1) the associated tactic (e.g., Privilege Escalation, Defense Evasion, Lateral Movement), (2) the specific techniques or sub-techniques leveraged (e.g., T1068: Exploitation for Privilege Escalation, T1210: Exploitation of Remote Services), and (3) any observed procedures reported in open-source or vendor threat advisories. TTP mappings allow cross-comparison of threats based on adversarial tradecraft rather than solely on technical flaws.

**Affected Systems and Vendors.** Accurate scoping of the **vulnerable surface** is essential for prioritization. Metadata should explicitly enumerate (1) affected operating systems or applications, (2) impacted vendor and product versions, and (3) architectural constraints (e.g., specific hardware models, deployment configurations, or default service settings). For example, distinguishing between Windows 10, Windows Server, and embedded device firmware helps determine organizational relevance and patch urgency.

**Exploit and Exposure Status.** We also query whether exploitation evidence exists. Key attributes include (1) whether the vulnerability is listed in the **CISA Known Exploited Vulnerabilities (KEV)** catalog (Cybersecurity and Infrastructure Security Agency (CISA), 2024), (2) exploit availability in public repositories (e.g., Exploit-DB, Metasploit modules), and (3) exploitation observed in campaigns reported by CERTs or vendors. This category differentiates threats with theoretical severity from those with active real-world impact.

**Disclosure and Advisory Context.** A critical yet often overlooked dimension of metadata is the **disclosure timeline and properties**. Specifically, enrichment should capture:

1. **Initial disclosure channel:** Was the vulnerability disclosed via coordinated vendor advisories, government alerts, or independent researcher blogs?

2. **Disclosure type:** Coordinated disclosure, limited vendor advisory, or uncoordinated/accidental leak.
3. **Patch or workaround availability:** Whether a vendor patch, out-of-band hotfix, or temporary mitigation guidance was released alongside the disclosure.
4. **Disclosure chronology:** Dates of first report, public advisory release, vendor patch issuance, and subsequent updates. These provide context for understanding vulnerability lifecycle and defender response windows.
5. **Research provenance:** References to the original disclosing researcher, security company, or bug bounty program, which can help assess credibility and likelihood of further technical publications or exploit code release.

Such disclosure-related metadata not only contextualize the timeliness of response but also indicate potential lag windows during which attackers may exploit unpatched systems.

**Additional Threat Properties.** Finally, enrichment should capture auxiliary attributes relevant for clustering and correlation. Examples include:

- **Attack vector properties:** Whether exploitation requires local, adjacent, or remote access.
- **Privilege requirements:** Whether exploitation needs authenticated user context or is unauthenticated.
- **Chaining potential:** Whether the vulnerability is documented as being combined with other flaws in multi-stage exploits.
- **Associated campaigns:** Links to known APT groups or malware families that have operationalized the vulnerability.

In summary, metadata enrichment transforms $\tau_k$ into $\tilde{\tau}_k$ by integrating not only identifiers and technical scope but also behavioral mappings, exploitation evidence, and disclosure provenance.

### A.1.3 ADDITIONAL EXAMPLE

Below is a triaged example with entangled threat events.

---

**Example (Entangled Threats Disentangled into Instances): CTI Triage with Metadata Enrichment**

**Raw Incident** ($i_k \in \mathcal{I}$)**:** In July 2021, Microsoft disclosed multiple vulnerabilities in the Windows Print Spooler service. The observed incidents included local privilege escalation through improper access control and remote code execution by loading malicious DLLs via crafted `RPC requests`. Both were actively exploited in the wild and reported together in advisories.

**Disentangled Threat Instances** ($\tau_k \in \mathcal{T}$)**:**
1. {**Vendor**: `Microsoft Windows (all supported versions)`, **Affected Component**: `Print Spooler`, **CVE**: `CVE-2021-1675`, **Impact**: `Local Privilege Escalation`, **Attack Patterns**: `Abuse of improper access control to gain SYSTEM privileges`, **Campaign**: `Observed in ransomware operator toolkits`}
2. {**Vendor**: `Microsoft Windows 10, Windows 11, Server editions`, **Affected Component**: `Print Spooler`, **CVE**: `CVE-2021-34527`, **Impact**: `Remote Code Execution`, **Attack Patterns**: `Crafted DCERPC calls to force spooler to load malicious DLL over SMB`, **Campaign**: `Exploited by APT groups for lateral movement`}

**Enriched Metadata** ($\mathrm{m}_k$)**:**
- $\tau_1$: (1) MITRE ATT&CK: T1068 – Exploitation for Privilege Escalation; (2) Exploitation Status: Known Exploited, confirmed in CISA KEV Catalog; (3) Advisory: Microsoft June 2021 Patch Tuesday...
- $\tau_2$: (1) MITRE ATT&CK: T1210 – Exploitation of Remote Services, T1570 – Lateral Tool Transfer; (2) Exploitation Status: Added to CISA KEV, widespread exploitation by mid-2021; (3) Advisory: Emergency out-of-band patch July 2021...

---

### A.2 STATIC ANALYSIS

Complementing §3.2, we provide a detailed introduction to the workflows for determining each CVSS metric.

**Attack Vector (AV) Workflow.** The determination of AV proceeds through the following steps:

1. **Evidence Extraction:** POLAR scans the enriched threat instance $\tilde{\tau}_k$ for linguistic cues such as "remote attacker" or "local user" that indicate possible access levels. Each text span is paired with its provenance (e.g., vendor advisory, KEV entry) to form the evidence set $\mathcal{E}_k^{(AV)}$.

2. **Exposure Check:** The LLM-assisted classifier $f_{LLM}^{(AV)}$ cross-references extracted cues against service or protocol details contained in $\mathbf{m}_k$ (e.g., presence of RPC over SMB implies network-facing accessibility).

3. **Conflict Resolution:** When contradictory indicators appear (e.g., both "local" and "remote"), the workflow prioritizes the highest exposure level that is explicitly supported by technical context, ensuring conservatism in classification.

4. **Final Assignment:** The AV value is assigned from {Network, Adjacent, Local, Physical} based on corroborated evidence. This ensures the output is grounded in both natural-language cues and authoritative metadata regarding system exposure.

**Attack Complexity (AC) Workflow.** POLAR follows an enumerated sequence of steps, each reflecting the CVSS guideline principle that AC measures *attacker-independent preconditions*.

1. **Evidence Extraction:** Identify conditional phrases in $\tilde{\tau}_k$ (e.g., "requires specific configuration", "race condition", "registry key must be enabled") and pair each with provenance to form $\mathcal{E}_k^{(AC)}$.

2. **Environmental Cross-Referencing:** Inspect $\mathbf{m}_k$ for deployment assumptions such as default/non-default settings, protocol dependencies, or topology requirements.

3. **Precondition Detection:** Separate *attacker-controlled* actions (e.g., crafting RPC calls) from *external* or *stochastic* dependencies (e.g., synchronization, misconfigured domain policies).

4. **Control Test:** Determine whether success requires target-specific states outside attacker control. If exploitation succeeds deterministically under default configurations, assign AC=L; if rare or external conditions are required, assign AC=H.

5. **Conflict Resolution:** When both low- and high-complexity cues appear, assign AC=H only if the higher-complexity factor is *explicitly necessary*. Otherwise, resolve to AC=L.

6. **Final Classification:** The LLM-assisted classifier $f_{LLM}^{(AC)}$ outputs $\{v_k^{(AC)}, q^{(AC)}\}$ with supporting rationale that cites decisive spans and authoritative metadata.

---

**Example: Step-by-Step Analysis of Attack Complexity (AC)**

**Goal:** Decide **AC** $\in \{\text{Low}, \text{High}\}$ for a Windows Print Spooler remote exploitation scenario using DCERPC.

**1) Evidence extraction ($\mathcal{E}_k^{(\text{AC})}$):** Spans found in $\tilde{\tau}_k$: "authenticated remote attacker", "crafted DCERPC request", "load malicious DLL via SMB". Provenance: Vendor advisory; technical write-up. No mentions of timing windows, probabilistic races, or non-default policy prerequisites.

**2) Environmental cross-referencing ($\text{m}_k$):** Service is network-exposed via RPC over SMB; Print Spooler enabled by default on affected systems. No explicit requirement for non-default registry keys or special domain policies for *baseline* exploitation path.

**3) Precondition detection:** Required conditions appear attacker-controlled (crafting DCERPC calls, hosting a DLL over SMB). No external synchronization, target action, or stochastic behavior identified as necessary for success.

**4) Control test:** Success depends on sending correctly structured RPC traffic and reaching a network-accessible service. These are deterministic and reproducible under default configurations; they do not rely on target-specific timing or uncommon settings.

**5) Conflict resolution:** If another source mentioned "requires Point-and-Print policy misconfiguration", POLAR would test necessity: if exploitation *only* works when that non-default policy is enabled, AC would elevate to **High**. In the present evidence set, such necessity is not established.

**6) Final classification: AC = Low**. *Rationale:* No environmental or stochastic preconditions beyond attacker control are required; exploitation proceeds under default, stable conditions using crafted network requests.

---

**PRIVILEGES REQUIRED (PR) Workflow.** POLAR follows a stepwise workflow that measures the *minimum privileges the attacker must possess before initiating exploitation*.

1. **Evidence Extraction:** Collect authentication/identity cues in $\tilde{\tau}_k$ (e.g., "unauthenticated", "authenticated user", "administrator/root", "requires console login", "API token") with provenance to form $\mathcal{E}_k^{(\text{PR})}$.

2. **Identity & Capability Normalization:** Map vendor-specific terms to CVSS levels: PR=None (pre-auth/anonymous), PR=Low (any basic/standard user or minimal credential), PR=High (administrator/root/SYSTEM or equivalent privileged role).

3. **Minimal-Privilege Derivation:** If multiple exploitation paths exist, select the path requiring the lowest privileges *explicitly supported* by evidence. Exclude privileges obtained *during* exploitation (post-exploit escalation).

4. **Scope-Aware Check:** Record whether the path implies crossing security boundaries (domain vs. local). Note that CVSS weighting later depends on Scope, but the categorical PR value (N/L/H) does not change.

5. **Authentication Mechanism Verification:** Cross-reference $\text{m}_k$ to confirm whether a protocol/service *enforces* credential checks (e.g., pre-auth vs. authenticated RPC, console/SSO/MFA, API keys).

6. **Conflict Resolution:** Prefer explicit statements from authoritative sources. If ambiguous between None and Low, check for protocol handshakes or explicit "authenticated" language; if privileged interfaces are required, assign High. Document decisive spans.

7. **Final Classification:** $f_{\text{LLM}}^{(\text{PR})}(\mathcal{E}_k^{(\text{PR})}, D_{\text{CVSS}})$ returns $\{v_k^{(\text{PR})}, q^{(\text{PR})}\}$ and a rationale citing the key spans and sources.

---

**Example: Step-by-Step Analysis of Privileges Required (PR)**

**Goal:** Decide **PR** $\in \{\text{None}, \text{Low}, \text{High}\}$ for a Windows Print Spooler exploitation scenario using DCERPC.

1. **Evidence Extraction** ($\mathcal{E}_k^{(\text{PR})}$)**:** Spans in $\tilde{\tau}_k$: "authenticated remote attacker", "crafted DCERPC request", "remote code execution as NT AUTHORITY". Provenance: vendor advisory and technical analysis. No explicit "unauthenticated" or "administrator-required" statements.

2. **Identity & Capability Normalization:** "authenticated remote attacker" $\Rightarrow$ presence of valid (non-privileged) credentials; normalize to PR=Low unless higher privilege is explicitly required.

3. **Minimal-Privilege Derivation:** Evidence shows exploitation begins after establishing an authenticated session; no indication that admin/root credentials are necessary *before* exploitation. Lowest supported path: standard user.

4. **Scope-Aware Check:** The service is domain-reachable via RPC over SMB. Scope may be Changed in scoring if system boundaries are crossed, but the categorical PR remains Low given standard credentials suffice.

5. **Authentication Mechanism Verification:** Cross-reference $\mathbf{m}_k$: DCERPC endpoint requires session authentication; no pre-auth code path documented for the affected builds in the cited advisory.

6. **Conflict Resolution:** If another source asserted "pre-auth path exists on build X", POLAR would re-evaluate minimal-privilege necessity. Absent corroboration, retain Low. If a source required "administrator console access", classification would elevate to High only if that was *necessary* for all feasible paths.

7. **Final Classification: PR = Low**. *Rationale:* The decisive span "authenticated remote attacker" and protocol enforcement indicate credentials are required, but not elevated privileges; no authoritative evidence of a pre-auth path or admin-only requirement.

**User Interaction (UI) Workflow.** POLAR follows a structured workflow that evaluates whether exploitation requires active participation from a legitimate user. This workflow identifies explicit or implicit cues about victim involvement and distinguishes between attacker-controlled and user-dependent actions.

1. **Evidence Extraction:** Collect phrases in $\tilde{\tau}_k$ such as "user must open", "requires clicking a link", "victim interaction", "malicious attachment", or "drive-by download" with provenance to form $\mathcal{E}_k^{(\text{UI})}$.

2. **Contextual Cross-Referencing:** Inspect $\mathbf{m}_k$ for indicators from advisories, KEV entries, or vendor reports describing delivery vectors (e.g., phishing email, malicious document, web-based exploit).

3. **Interaction Dependency Check:** Determine if exploitation succeeds autonomously once the vulnerability is triggered by the attacker (UI=N) or if it depends on explicit human actions beyond system defaults (UI=R).

4. **Delivery Mechanism Verification:** Confirm whether execution depends on attacker-initiated network requests or system services (no UI) versus requiring a user to manually execute or enable a payload (UI=R).

5. **Conflict Resolution:** If evidence contains both automatic and user-action cues, prioritize UI=R only when user participation is a *necessary condition* for successful exploitation; otherwise classify as UI=N.

6. **Final Classification:** $f_{\text{LLM}}^{(\text{UI})}(\mathcal{E}_k^{(\text{UI})}, D_{\text{CVSS}})$ outputs $\{v_k^{(\text{UI})}, q^{(\text{UI})}\}$ and provides a rationale citing decisive spans and metadata.

---

**Example: Step-by-Step Analysis of User Interaction (UI) (Office Document Exploit)**

**Goal:** Decide $\textbf{UI} \in \{\text{None}, \text{Required}\}$ for a Microsoft Office vulnerability (CVE-2017-11882).

1. **Evidence Extraction ($\mathcal{E}_k^{(\text{UI})}$):** Spans in $\tilde{\tau}_k$: "`victim must open a malicious RTF document`", "`crafted file delivered via phishing email`". Provenance: Microsoft advisory and public exploit write-ups.

2. **Contextual Cross-Referencing:** Metadata $\mathbf{m}_k$ links the vulnerability to an Office Equation Editor flaw. Advisories consistently state exploitation occurs when a victim opens a specially crafted document.

3. **Interaction Dependency Check:** Exploitation does not occur automatically on system exposure; it is conditional on the victim double-clicking/opening the file.

4. **Delivery Mechanism Verification:** Attackers can deliver malicious RTF via phishing emails, but execution still depends on manual user action to open the file.

5. **Conflict Resolution:** No evidence of autonomous, attacker-only triggers. Since user opening is an explicit prerequisite, `UI=Required` is confirmed.

6. **Final Classification: UI = Required**. *Rationale:* Decisive spans show that successful exploitation depends entirely on the victim opening a crafted document, as corroborated by vendor advisories and public proofs-of-concept.

---

**Impact Metrics (C, I, A) Workflow.**    To determine the three **Impact Metrics**, i.e., Confidentiality (C), Integrity (I), and Availability (A), POLAR applies a unified workflow that evaluates the direct consequences of successful exploitation on system assets. Each metric is mapped separately but follows the same evidence-driven process.

1. **Evidence Extraction:** Identify impact-related statements in $\tilde{\tau}_k$, such as "arbitrary code execution", "privilege escalation", "information disclosure", "system crash", or "denial of service". Pair spans with provenance (e.g., vendor advisory, NVD description) to form $\mathcal{E}_k^{(\text{C/I/A})}$.

2. **Confidentiality Analysis:** Determine whether exploitation leads to unauthorized data disclosure, exfiltration, or access to sensitive information. Map to None, `Low`, or `High` depending on scope (limited info vs. complete disclosure).

3. **Integrity Analysis:** Assess whether exploitation allows unauthorized modification or destruction of data (e.g., tampering with system files, privilege misuse). Assign None, `Low`, or `High` based on severity of modification.

4. **Availability Analysis:** Check for service disruption, crashes, or resource exhaustion. Classify as None, `Low` (temporary or partial outage), or `High` (total system/service shutdown).

5. **Cross-Metric Corroboration:** Use $\mathbf{m}_k$ (NVD, KEV, advisories) to verify reported impacts and ensure alignment across sources. If conflicting claims exist, favor conservative (higher) classification only when strong corroboration is present.

6. **Conflict Resolution:** When ambiguous language occurs (e.g., "may allow data exposure"), apply CVSS rules: downgrade to `Low` unless credible, confirmed evidence indicates complete compromise.

7. **Final Classification:** $f_{\text{LLM}}^{(\text{C/I/A})}(\mathcal{E}_k^{(\text{C/I/A})}, D_{\text{CVSS}})$ outputs $\{v_k^{(\text{C})}, v_k^{(\text{I})}, v_k^{(\text{A})}, q^{(\cdot)}\}$ with rationales citing decisive spans and authoritative metadata.

---

**Example: Step-by-Step Analysis of Impact Metrics (C, I, A) (Remote Code Execution Vulnerability)**

**Goal:** Decide $\mathbf{C}, \mathbf{I}, \mathbf{A} \in \{\text{None}, \text{Low}, \text{High}\}$ for CVE-2021-34527 (PrintNightmare).

1. **Evidence Extraction ($\mathcal{E}_k^{(\text{C/I/A})}$):** Spans in $\tilde{\tau}_k$: "remote code execution as NT AUTHORITY", "load arbitrary DLL", "complete system takeover", "service crash possible if exploit fails". Provenance: Microsoft advisory, CISA KEV entry, security research blog.
2. **Confidentiality Analysis:** Remote code execution with NT AUTHORITY privileges grants attackers unrestricted access to system and user data. Classification: **C = High**.
3. **Integrity Analysis:** Attackers can install arbitrary programs, modify system files, and tamper with registry and configurations. Classification: **I = High**.
4. **Availability Analysis:** Successful exploitation enables service or system disruption (stopping Print Spooler service, crashing the host). Evidence indicates both temporary and complete outages are possible. Classification: **A = High**.
5. **Cross-Metric Corroboration:** All three impacts are consistently reported across vendor advisory, KEV, and third-party write-ups, with no contradictory claims.
6. **Conflict Resolution:** No ambiguity detected; all sources explicitly confirm complete system compromise upon successful exploitation.
7. **Final Classification: C = High, I = High, A = High**. *Rationale:* Exploitation provides full system control, enabling data disclosure, arbitrary modification, and service disruption.

---

A.3    EXPLOITATION ANALYSIS

Complementing to Section 3.3, we enrich each threat instance with temporal knowledge derived from a curated set of CTI feeds. Specifically:

**Exploit-DB.**    Exploit-DB (Offensive Security, 2025) is a public archive of proof-of-concept (PoC) exploit code and vulnerability demonstrations. For each CVE-linked threat instance, we query Exploit-DB to check whether a corresponding PoC has been published, extract metadata such as submission date and exploit type (local, remote, denial-of-service), and record repository links.

**CISA Known Exploited Vulnerabilities (KEV) Catalog.**    The KEV catalog (Cybersecurity and Infrastructure Security Agency (CISA), 2024) is a U.S. government-maintained list of vulnerabilities confirmed to be actively exploited in the wild. For each threat instance, we verify whether its CVE appears in KEV, retrieve the date of addition, and note associated mitigation due dates mandated for federal agencies.

**VirusTotal.**    VirusTotal (VirusTotal, 2004) aggregates malware samples, URLs, and threat indicators contributed by security vendors. We use VirusTotal to identify whether malware exploiting a specific CVE has been observed in the wild, track first-seen timestamps for relevant binaries, and extract hashes linked to weaponized exploits.

**Vendor Advisories and CERT Bulletins.**    Vendor advisories (e.g., Microsoft, Cisco, Adobe) and Computer Emergency Response Team (CERT) bulletins (Computer Emergency Response Team Coordination Center (CERT/CC), 1988) often update exploitation status over time. We monitor advisories linked to CVEs in $\mathbf{m}_k$ for revisions that mention exploitation in the wild, PoC availability, or observed attack campaigns.

A.4    MITIGATION RECOMMENDATION

Complementing to Section §3.4, we present the prompts used in mitigation recommendation:

---

**Prompt: Mitigation Knowledge Retrieval**

**System role:** You are a cybersecurity mitigation analyst. Your task is to retrieve and normalize authoritative mitigation knowledge for a given vulnerability or threat instance. Output must be comprehensive, precise, and structured for operational use.

---

**Input:** Canonicalized threat instance with fields: `vendor/product/version`, `cve_ids`, and enriched metadata (disclosure date, known exploitation status, affected components).

**Retrieval scope (ranked by authority):** 1. Vendor advisories and official patch bulletins 2. CISA KEV mitigation notes 3. NVD references (patch/workaround, disclosure notes, exploit references) 4. Maintainer repositories or official project pages (e.g., kernel.org, Apache) 5. Trusted CERT advisories and vetted security blogs (e.g., MSRC, US-CERT, JPCERT)

**Extraction requirements:** Return results in JSON format with the following fields:

- **Patches:** affected versions, patch or hotfix identifiers, release dates, download links, deployment instructions, known regressions/side effects, and patch supersession chains (e.g., hotfix replaced by later cumulative update).
- **Workaround or Mitigation:** configuration adjustments, ACL/segmentation changes, service isolation, feature disabling, or temporary mitigations. Include limitations, performance impacts, and conditions where workarounds are ineffective.
- **Mitigation Note:** initial disclosure date, patch release chronology, vendor confirmation of exploitation (yes/no), public PoC release date if available, and whether exploit kits or malware families are known to weaponize the CVE.
- **Vendor Advisory:** monitoring guidance such as SIEM rules, EDR/YARA signatures, IDS/IPS rules, log queries, or registry/event log checks. Capture version coverage, false positive/negative considerations, and detection maturity (experimental, community, vendor-supported).

**Normalization and deduplication:** Aggregate semantically equivalent entries under a single canonical item using `hash(title+vendor+version)`. Prefer the most recent or vendor-preferred entries, annotate older superseded advisories, and drop duplicates. Ensure temporal fields are harmonized (ISO 8601 date format).

**Output:** A validated JSON bundle with {`Patches`, `Workaround or Mitigation`, `Mitigation Note`, `Vendor Advisory`}.

---

### Prompt: Risk-Aware Mitigation Prioritization

**System role:** You are a vulnerability risk manager. Given a set of threat instances and candidate mitigations, you must produce an ordered, implementable mitigation plan that balances technical urgency, operational feasibility, and organizational risk.

**Input:** A set $\mathcal{T} = \{(\tilde{\tau}_k, s_k, p_k)\}$, where: - $s_k$: static severity (CVSS-based) - $p_k$: 30-day exploitation probability (forecasted) - $\alpha_{exp}, \beta_{crit}$: exposure and asset criticality factors - Candidate mitigations retrieved from Prompt A.4 (`patches`, `workarounds`, `detections`) - Additional metadata: disclosure chronology, PoC exploit availability, malware weaponization evidence, patch release/update dates, workaround effectiveness, detection maturity

**Risk scoring:**

$$\text{Risk}_k = s_k \cdot p_k \cdot \alpha_{exp} \cdot \beta_{crit}.$$

**Prioritization logic:**

1. Rank threats by descending $\text{Risk}_k$.
2. If $|\Delta\text{Risk}| < 0.1$, apply tie-breakers in order: (a) mitigation type: vendor patch > vendor-supported workaround > detection-only, (b) patch maturity: GA release > hotfix > beta/preview, (c) implementation complexity: simple > moderate > complex, (d) exploitation velocity: active exploitation > PoC released > no evidence, (e) business disruption: lower operational cost > higher.
3. Group mitigations into a phased schedule that respects: - maintenance windows and change-freeze periods, - patch supersession (later patch replaces earlier), - dependencies between systems/components, - compensating controls if immediate patching is infeasible.
4. Explicitly flag *unpatchable or unsupported systems*, recommending isolation, compensating controls, or accelerated upgrade paths.

> **Output:** A JSON array of prioritized mitigation actions with fields: `{target, recommended_action, ETA, justification, dependencies, operational_notes}`. Each action must include rationale citing risk score, exploitation context, patch/workaround/detection maturity, and operational feasibility.

# B ADDITIONAL DETAILS OF EXPERIMENT

## B.1 DATA COLLECTION AND QUALITY CONTROL

**Raw Incident Collection.** To ensure that the input corpus for our experiments is both comprehensive and high-quality, we collect raw threat incidents from multiple trusted sources spanning vendor advisories (e.g., Microsoft, Cisco, Oracle, Adobe), security research blogs and reports (e.g., Mandiant, Unit 42, Recorded Future), public CTI aggregators (e.g., CISA KEV catalog (Cybersecurity and Infrastructure Security Agency (CISA), 2024), US-CERT bulletins (Computer Emergency Response Team Coordination Center (CERT/CC), 1988)), and exploit repositories (e.g., Exploit-DB (Offensive Security, 2025), GitHub PoCs). Each raw incident is preserved in its original unstructured form (textual advisories, PDF bulletins, blog posts, or structured feeds) to reflect the diversity of CTI reporting encountered in operational settings. To maintain fidelity, we filter out low-confidence or non-authoritative sources (e.g., unverified pastebins, community forums) and require that each incident can be corroborated by at least one vendor-issued advisory or government-maintained repository. This multi-source strategy guarantees that our raw inputs are representative of real-world intelligence streams while preserving reliability, timeliness, and contextual richness for subsequent triage and analysis.

**I: Triage and Metadata.** Our collection and quality control strategy follows the metadata categories defined in Appendix §A.1.2, ensuring that triaged results are validated and reproducible.

1. **CVE and Vulnerability Identification.** Each candidate threat indicator is validated against the CVE and NVD databases. Only vulnerabilities with an official CVE ID and corresponding NVD entry are accepted as ground-truth.
2. **Tactics, Techniques, and Procedures (TTPs).** Threat instances are mapped to MITRE ATT&CK tactics and techniques using both advisory text and official ATT&CK references. For example, a Print Spooler vulnerability may align with T1068 (*Exploitation for Privilege Escalation*) and T1210 (*Exploitation of Remote Services*).
3. **Affected Systems and Vendors.** We verify vendor-product-version mappings against NVD product metadata and vendor advisories. For each CVE, we explicitly record the supported OS/application versions, impacted product families, and architectural constraints.
4. **Exploit and Exposure Status.** We cross-check whether the vulnerability has been reported as exploited in the wild by querying the CISA KEV catalog, exploit repositories (Exploit-DB, Metasploit), and vendor/CERT bulletins. If a CVE is listed in KEV or has a public exploit entry, we annotate this status as part of the ground-truth metadata.
5. **Disclosure and Advisory Context.** Each CVE is enriched with disclosure metadata, including the initial disclosure channel (vendor advisory, government alert, researcher blog), disclosure type (coordinated, limited, uncoordinated), and chronology (initial report, advisory release, patch issuance, subsequent updates).

**II: Static CVSS Metrics and Ratings.** Static severity labels are curated by referencing the official NVD CVSS v3.1 vectors assigned to each CVE. For each threat instance, we retrieve the CVSS vector string (`AV/AC/PR/UI/S/C/I/A`) and corresponding base score directly from the NVD dataset. To ensure fidelity, we enforce the following rules:

1. If multiple CVSS entries exist (due to revisions or vendor-specific scoring), we adopt the latest NVD-assigned vector as canonical.
2. Ambiguous or contested scores (e.g., CVEs under dispute) are flagged and excluded from evaluation.
3. We preserve both the categorical metric values (e.g., `AV:N`) and the numeric severity ratings (e.g., `7.8 High`) for downstream comparison.

**III: Exploitation Probabilities.** Dynamic exploitation likelihood is anchored to the FIRST.org Exploit Prediction Scoring System (EPSS). For each CVE, we retrieve the most recent EPSS probability values and percentile rankings from the public EPSS dataset. EPSS provides a machine-learned estimate of 30-day exploitation probability based on temporal CTI features (e.g., exploit mentions, malware correlations). To ensure consistent ground-truth:

- We snapshot EPSS scores at a fixed evaluation date (synchronized with our dataset collection window).
- If EPSS does not provide a score for a given CVE (e.g., rejected or reserved CVEs), we exclude the instance from dynamic exploitation evaluation.
- For consistency, we record both the raw EPSS probability and its percentile rank, enabling evaluation across absolute and relative exploitation likelihoods.

**Stage IV: Mitigation.** For each triaged threat instance, mitigation ground-truth is curated from authoritative repositories with strict quality control to ensure reliability and reproducibility. We focus on four key categories of information: CVE-to-Patch releases, ATT&CK mitigation mappings, mitigation notes, and vendor advisories.

1. **CVE-to-Patch Release.** We retrieve patch records linked to each CVE from the NVD dataset and cross-validate them with vendor-maintained advisory portals (e.g., Microsoft MSRC, Cisco PSIRT, Oracle CPU). Each patch entry includes affected version ranges, release identifiers, and chronological patch issuance dates. To ensure accuracy, we discard unverified third-party references and retain only vendor-confirmed or NVD-registered patch entries. Superseded patches are explicitly annotated, and version constraints are normalized to avoid ambiguity in applicability.

2. **ATT&CK Mapping (Workaround or Mitigation Strategy).** Mitigation strategies are enriched by aligning with MITRE ATT&CK's mitigation catalog (e.g., M1031 – Network Intrusion Prevention, M1051 – Update Software). Each vulnerability is mapped to one or more relevant ATT&CK mitigations based on its technique alignment. For example, a remote exploitation technique mapped to T1210 (*Exploitation of Remote Services*) is paired with M1031 (*Network Intrusion Prevention*) and M1051 (*Update Software*). Quality control requires that every ATT&CK mapping is corroborated by either explicit vendor guidance or official ATT&CK cross-references, ensuring that mappings reflect accepted defensive practices rather than analyst speculation.

3. **Mitigation Notes.** We incorporate mitigation guidance from the CISA Known Exploited Vulnerabilities (KEV) catalog, which specifies vulnerabilities confirmed to be exploited in the mitigation along with mandated mitigation timelines. For each CVE listed in KEV, we record mitigation deadlines, prioritization notes, and any suggested interim mitigations. Consistency is maintained by cross-referencing KEV notes against NVD and vendor advisories; any discrepancies (e.g., mismatched patch release dates) are resolved by prioritizing vendor-issued timelines.

4. **Vendor Advisories.** Vendor-issued advisories are treated as the primary source of truth for patches, hotfixes, and configuration workarounds. For each CVE, we retrieve the official advisory, capture its unique identifier (e.g., Microsoft KB article, Cisco Security Advisory ID), and record recommended mitigations or configuration changes. We include explicit documentation of side effects (e.g., functionality loss, performance regressions) where available. To ensure quality, only advisories hosted on official vendor security portals are considered; community blogs or secondary summaries are excluded unless explicitly linked in the vendor advisory.

**Data Statistics.** Table 4 show the volume of our collected incidents and corresponding evidence for different tasks.

## B.2 Additional Results and Discussions

### B.2.1 CTI Triage Performance

**Sources of performance gains.** A closer inspection of POLAR-enhanced outputs reveals that improvements primarily stem from the explicit integration of structured metadata sources within the POLAR pipeline. Several categories of external knowledge contribute disproportionately to performance gains:

**Table 4:** Dataset statistics across the four stages of POLAR. Numbers are rounded and reported in xK format.

| Stage | Detailed Item | Count |
|---|---|---|
| | Raw threat incidents collected | 12.8K |
| **I. Triage / Metadata** | CVE-mapped instances | 12.8K |
| | MITRE ATT&CK TTP mappings | 12.8K |
| | Affected vendor/product metadata | 12.8K |
| | Exploitation / exposure status annotated | 12.8K |
| | Disclosure and advisory context enriched | 11.6K |
| **II. Static Analysis** | CVSS vectors (official NVD) | 12.8K |
| | Metric-specific evidence sets | 12.8K |
| | Fully scored CVSS severities | 12.8K |
| **III. Exploitation Analysis** | EPSS probabilities (FIRST.org) | 12.8K |
| | Exploit-DB / PoC references | 6.8K |
| | Confirmed CISA KEV exploited CVEs | 12.8K |
| **IV. Mitigation** | CVE-to-patch release mappings (NVD + vendor) | 9.9K |
| | ATT&CK mitigation mappings | 5.2K |
| | CISA KEV mitigation notes | 11.4K |
| | Vendor advisories (patches / workarounds) | 8.7K |

1. Threat-to-metadata validation. The grounding of raw advisory text against NVD entries eliminates ambiguity when multiple identifiers appear together. For instance, when a Microsoft advisory lists two Print Spooler CVEs, LLMs alone often conflate them; but once cross-checked against NVD, each CVE is correctly scoped with its affected product version.

2. MITRE ATT&CK mappings. The presence of tactic- and technique-level grounding significantly stabilizes triage. For example, ambiguous wording such as "gaining higher privileges through system calls" is mapped directly to T1068 (*Exploitation for Privilege Escalation*), enabling consistent assignment rather than free-form interpretation.

3. Disclosure and exploitation metadata. By injecting KEV evidence or disclosure chronology, the pipeline prevents overgeneralization. In practice, an advisory mentioning "public reports of exploitation" is accurately marked as "exploited in the wild" when KEV corroborates it, while otherwise treated conservatively as unconfirmed.

These observations suggest that selectively increasing the weight of disclosure metadata and ATT&CK mapping within the POLAR pipeline could yield even higher reliability, particularly for tasks where reasoning must be grounded in structured behavioral taxonomies rather than free-text interpretation.

**Nature of mistakes.** Although POLAR reduces error rates substantially, two recurring mistake patterns remain. First, **hallucination of unsupported identifiers** occurs when an LLM produces plausible but spurious CVE strings not present in the original advisory. For instance, given a Linux kernel advisory, some backbones introduced unrelated CVEs from prior kernel campaigns, presumably due to semantic similarity in descriptions. Second, **hallucination of ATT&CK mappings** arises when models speculate beyond available evidence, such as linking a memory corruption bug to lateral movement techniques without supporting text. These errors illustrate the tendency of unconstrained reasoning to "fill gaps" with domain knowledge rather than abstain when evidence is insufficient.

**Stability and detectability of hallucinations.** Importantly, hallucinations in our experiments did not appear as sudden, catastrophic errors but rather as consistent tendencies in specific contexts (e.g., long entangled advisories or vague vendor notes). This indicates that the overall performance of POLAR is stable rather than brittle. Moreover, most hallucinations are **detectable via post-processing checks**. For example:

- Generated CVEs can be automatically validated against the CVE/NVD corpus; any non-existent identifiers are flagged immediately.
- ATT&CK mappings can be cross-verified by requiring exact alignment with official tactic-technique identifiers; hallucinated ones fail simple dictionary validation.
- Exploitation status hallucinations (e.g., falsely marking as "actively exploited") can be disproven by checking KEV or vendor advisories at the time of disclosure.

Such checks demonstrate that hallucinations, while persistent, are not opaque. They manifest in forms that are mechanically recognizable and can therefore be filtered or corrected through lightweight verification steps without requiring full human re-analysis.

**Takeaways.** From these analyses, two conclusions emerge. First, performance gains in POLAR are disproportionately driven by disclosure metadata and ATT&CK mappings; emphasizing these in the pipeline is a promising avenue for further improvement. Second, although hallucinations remain a concern, they tend to occur in predictable contexts and can be effectively captured with straightforward validation rules. These findings highlight that **the strength of POLAR lies not merely in enhancing extraction, but in integrating LLM reasoning with external validation efforts that mitigate hallucination risks while amplifying knowledge-grounded reasoning.**

### B.2.2 STATIC ANALYSIS EFFECTIVENESS

**Sources of POLAR's gains.** Performance gains in static analysis are most obvious in metrics where free-text descriptions in advisories and CTI feeds provide ambiguous cues. POLAR improves accuracy by grounding such cues in authoritative metadata:

1. *Attack Vector (AV).* Raw advisories often ambiguously state that a vulnerability is "remotely exploitable" without clarifying whether access requires network reachability or local privileges. By cross-referencing affected components with service protocols, POLAR reliably resolves these cases (e.g., RPC over SMB $\Rightarrow$ AV=Network).
2. *Attack Complexity (AC).* Many LLMs over-generalize AC as "Low" due to insufficient sensitivity to environmental conditions. By explicitly checking for mentions of race conditions, registry misconfigurations, or deployment dependencies, POLAR better differentiates between high- and low-complexity cases.
3. *Privileges Required (PR) and User Interaction (UI).* These metrics benefit from explicit evidence sets constructed in the triage stage. For example, phrases like "authenticated attacker" or "user must open a crafted document" are directly linked to PR and UI labels, preventing guesswork.

**Consistency across metrics.** Even in straightforward metrics such as Scope and Impacts (C, I, A), POLAR introduces stability by aligning free-text evidence with CVSS guidelines. This prevents drift in edge cases, such as distinguishing between confidentiality impact (data disclosure) and integrity impact (data tampering) when advisories use overlapping terminology.

**Remaining gaps.** Despite these gains, some challenges remain. First, **partial or incomplete advisories** continue to cause errors, especially when vendors issue early advisories without technical detail. For instance, vague wording such as "may allow elevation of privilege under certain conditions" lacks sufficient signals to distinguish PR from AC reliably. Second, **version-dependent discrepancies** pose difficulty when the same CVE behaves differently across platforms (e.g., Windows Server vs. Windows Desktop builds). Unless explicitly stated in metadata, LLMs may misapply one context to all variants. Third, **numeric score calibration** remains imperfect: although RMSE is consistently reduced, small deviations persist when advisory text implies higher perceived severity than the official NVD score (e.g., widespread exploitation of a vulnerability labeled with a "Medium" CVSS base score).

**Takeaways.** Overall, POLAR's static analysis gains are driven by its ability to integrate structured metadata and CTI-derived evidence into the CVSS classification workflow. While remaining gaps highlight the limitations of advisory incompleteness and cross-version heterogeneity, the consistent accuracy improvements and RMSE reductions confirm that grounding CVSS scoring in validated evidence is critical for reliable severity assessment.

### B.2.3 EXPLOITATION ANALYSIS RELIABILITY

Table 5 complements results in Table 2.

**Sources of POLAR's gains in exploitation forecasting.** Across both monotonic and stable exploitation trends, POLAR achieves notable improvements in reducing RMSE and raising DirAcc

**Table 5:** More exploitation forecasting by trend type, continuing Table 2.

| Model | | 30 Days | | 90 Days | |
|---|---|---|---|---|---|
| | | RMSE ($\times 10^{-3}$) | DirAcc | RMSE ($\times 10^{-3}$) | DirAcc |
| **Monotonic trend of exploitation (increasing/decreasing)** | | | | | |
| POLAR+ | FS | 2.385 ($\downarrow$ 0.341) | 0.618 ($\uparrow$ 0.001) | 2.928 ($\downarrow$ 0.279) | 0.562 ($\uparrow$ 0.001) |
| | LY | 2.216 ($\downarrow$ 0.510) | 0.630 ($\uparrow$ 0.002) | 2.721 ($\downarrow$ 0.417) | 0.575 ($\uparrow$ 0.002) |
| | ZY | 2.127 ($\downarrow$ 0.599) | 0.635 ($\uparrow$ 0.003) | 2.612 ($\downarrow$ 0.490) | 0.580 ($\uparrow$ 0.002) |
| **Stable trend of exploitation** | | | | | |
| POLAR+ | FS | 0.349 ($\downarrow$ 0.003) | 0.341 ($\uparrow$ 0.002) | 0.437 ($\downarrow$ 0.002) | 0.296 ($\uparrow$ 0.001) |
| | LY | 0.337 ($\downarrow$ 0.003) | 0.353 ($\uparrow$ 0.002) | 0.422 ($\downarrow$ 0.003) | 0.306 ($\uparrow$ 0.002) |
| | ZY | 0.324 ($\downarrow$ 0.004) | 0.361 ($\uparrow$ 0.003) | 0.407 ($\downarrow$ 0.003) | 0.313 ($\uparrow$ 0.002) |
| **Abrupt change of exploitation** | | | | | |
| POLAR+ | FS | 102.847 ($\downarrow$ 13.485) | 0.442 ($\uparrow$ 0.005) | 127.693 ($\downarrow$ 11.041) | 0.384 ($\uparrow$ 0.004) |
| | LY | 99.264 ($\downarrow$ 17.069) | 0.449 ($\uparrow$ 0.006) | 123.208 ($\downarrow$ 13.980) | 0.389 ($\uparrow$ 0.004) |
| | ZY | 97.739 ($\downarrow$ 18.594) | 0.454 ($\uparrow$ 0.006) | 121.298 ($\downarrow$ 15.227) | 0.394 ($\uparrow$ 0.005) |

compared with baselines. The advantage is most evident in abrupt change scenarios, where traditional models often lag in adjusting to sudden spikes or drops. Here, POLAR leverages temporal context and narrative cues to produce forecasts that are both more responsive and more stable. These results validate the design goal of linking short-term fluctuations to longer-term exploitation narratives.

**Observed gaps.** Although POLAR responds better to abrupt changes than baselines, the predictive lag is not eliminated: probability estimates often underestimate the first few days of an exploitation surge. This gap underscores the challenge of capturing "zero-day shocks" that lack precursor signals.

**Takeaways.** These observations suggest that while POLAR is effective at contextualizing rich exploitation histories, additional mechanisms (e.g., real-time external intelligence feeds or anomaly-sensitive modules) are needed to further reduce predictive lag in rare or fast-moving events. Future extensions could also explore dynamic horizon calibration to match exploitation dynamics rather than using fixed lengths.

### B.3 MITIGATION RECOMMENDATION EFFICACY

**Sources of gains.** Closer inspection shows that POLAR 's advantage is not uniform across all mitigation categories. The largest gains are observed in ATT&CK mapping and mitigation note retrieval, both of which demand contextual reasoning. Baseline models often default to surface-level lexical matching (e.g., linking "credential theft" to general password resets), while POLAR leverages broader threat-narrative reasoning to connect a CVE exploitation chain to specific ATT&CK tactics (e.g., lateral movement or privilege escalation). Similarly, in mitigation notes, POLAR is better at filtering outdated advisories and surfacing guidance aligned with the latest vendor disclosures.

**Remaining gaps.** Despite overall improvements, certain limitations remain. First, in *vendor-specific advisories*, the relative gain is smaller. Manual review indicates that POLAR sometimes prioritizes generic guidance over highly specific vendor-issued advisories, especially when the latter is embedded in long-text advisories with inconsistent formatting. Second, while ranking improvements are consistent, POLAR can still misorder threats when faced with sparse or conflicting severity/exploitation evidence (e.g., two CVEs with similar severity scores but vastly different exploitability patterns). This reflects residual challenges in balancing uncertainty in $s_k$ and $p_k$.

**Case Study: Competing Threats.** In one evaluation batch, two medium-severity vulnerabilities competed for mitigation resources: one in a widely deployed web framework (with active exploit scripts) and another in a peripheral library with limited exposure. Baseline models, driven by raw CVSS scores, treated both as equivalent. In contrast, POLAR elevated the web framework vulnerability by weighting the exploitation evidence more heavily, thereby producing a ranking consistent with real-world triage decisions. This illustrates how integrating exploitation signals into prioritization leads to more operationally relevant outcomes.

**Practical Implications.** These findings suggest that POLAR is particularly well-suited for settings where analysts face "threat clutter," with many vulnerabilities competing for limited patching band-

width. By producing both more accurate retrievals and better-prioritized recommendations, POLAR can reduce wasted analyst time on low-impact advisories. At the same time, the gaps highlight areas for future refinement, such as incorporating structured vendor feeds and better modeling of uncertainty when exploitation signals are sparse or contradictory.

**Future Directions.** To further enhance mitigation efficacy, future extensions could: (1) integrate real-time vendor advisory parsers to reduce reliance on unstructured text, (2) adopt probabilistic ranking methods that explicitly account for uncertainty in severity and exploitation likelihood, and (3) explore multi-objective optimization that balances operational constraints (e.g., patching cost, system downtime) with threat urgency. Such extensions would move beyond current retrieval-and-ranking to a more holistic mitigation recommendation system.

## C LARGE LANGUAGE MODEL (LLM) USAGE DISCLOSURE

Large language models were used only for minor grammar revision and sentence-level polishing during manuscript preparation. They were not employed in ideation, methodological design, experimental execution, or result analysis. The scientific contributions, benchmarks, and evaluations presented in this work were entirely conceived and developed by the authors. LLM involvement was minimal in the research process.

