# OpenReview forum: "Polar: Automating Cyber Threat Prioritization through LLM-Powered Assessment"
_ICLR.cc/2026/Conference — ICLR 2026 Conference Withdrawn Submission_

### Official Review · Reviewer_fmQr · 2025-10-26

**Soundness:** 2
**Presentation:** 2
**Contribution:** 1
**Rating:** 2
**Confidence:** 4

**Summary:**

This paper presents POLAR which is a framework that uses large language models to help prioritize cyber threats automatically. The motivation is simple: there are too many vulnerabilities nowadays (over 11,000 new ones just in 2024), and security teams cannot handle all of them at once.

POLAR works in four stages: 1) it extracts threat information from security reports and enriches it with data from databases like NVD and MITRE ATT&CK. 2) it calculates CVSS severity scores using LLM reasoning. 3) it predicts if the vulnerability will be exploited soon by looking at historical patterns. 4) it recommends which threats to fix first and how to fix them.

The interesting thing is that the authors don't fine-tune the LLMs. They just use existing models like GPT or Claude with carefully designed prompts and connect them to security databases. They tested this on around 13K real vulnerabilities using 9 different LLM models, and POLAR consistently performs better than using the base models alone. The improvements are especially good when detecting sudden exploitation trends.

Overall, the paper tries to bridge the gap between automated tools and what security analysts actually need in practice.

**Strengths:**

S1. The paper tackles a critical and timely challenge in cybersecurity - the need to prioritize threats effectively amid exponentially growing vulnerability reports (11,000+ in 2024).

S2. The paper covers the entire workflow from reading threat reports to recommending actions, not just one isolated task. This end-to-end approach is more useful for real security teams than papers that only focus on one piece of the puzzle.

S3. The authors tested 13K real vulnerabilities using 9 different LLMs and ground truth from trusted sources (NVD, CISA, etc.). Table 2 shows different use trend types that are adaptable to different scenarios.

**Weaknesses:**

W1. The core contribution is primarily an engineering framework that applies existing LLMs through carefully designed prompts and retrieval augmentation. There's no novel architectural contribution, training methodology, or algorithmic innovation. The paper essentially uses LLMs as better text processors with access to structured databases. The "reasoning" is achieved through prompt engineering, which is well-established. No new model architecture, loss function, or training approach is proposed.

W2. POLAR's performance heavily relies on the accuracy, completeness, and timeliness of the external vulnerability databases and CTI feeds it consumes. Errors or delays in these sources could propagate through the system.

W2. All evaluations on known CVEs. But real threats include reports without CVE numbers, like misconfigurations, phishing campaigns, and zero-days before disclosure. The paper doesn’t address how Polar would handle these. So, the scope feels narrow for a solution that claims to do “cyber threat prioritization.”

W3. The evaluation primarily compares POLAR (using an LLM backbone) against the backbone LLM itself or other LLM agents. A comparison against established, non-LLM-based commercial or open-source threat prioritization tools/platforms (e.g., commercial SIEM/SOAR platforms, dedicated vulnerability management tools) would strengthen the claims of practical superiority.

W4. The paper focuses on offline evaluation metrics. Assessing the true real-world impact (e.g., reduction in analyst workload, faster patching of critical vulnerabilities, actual prevention of breaches) and computational overhead in a live environment remains an open question.

W5. The paper doesn't provide ablation studies to understand which components contribute most to performance.

**Questions:**

Q1. How does POLAR handle conflicting information potentially present in different CTI feeds or databases regarding a single threat (e.g., differing CVSS scores, conflicting exploitation reports)?

Q2. What is the computational cost (e.g., latency, API calls/token usage if applicable) of running the full POLAR pipeline for a batch of incidents compared to baseline LLM inference or typical analyst workflows?

Q3. Can you provide direct comparisons with EPSS and commercial vulnerability management systems?

Q4. Can you provide complete ablation studies showing the contribution of each component?

---

### Official Review · Reviewer_RXYc · 2025-10-30

**Soundness:** 2
**Presentation:** 3
**Contribution:** 2
**Rating:** 4
**Confidence:** 2

**Summary:**

The paper describes a threat prioritization mechanism leveraging LLM’s reasoning capabilities, and compare its performance with traditional rule-based or supervised machine learning approaches. Specifically, the proposed approach distangles threat instances, contextualizes it using external data resources, and scores its threat severity. Based on the scores, actionable mitigation strategies are recommended using prompts to retrieve these strategies from authoritative knowledge bases. The evaluation on numerous threat incidences demonstrate that the use of proposed approach with general-purpose and cyber-security specialized agents outperforms compared to its native scores.

**Strengths:**

- The paper considers the cyber threat prioritization problem as a multi-stage mechanism rather than a single scoring approach, using narrative-based reasoning thereby showing novelty in its contributions to cybersecurity defense workflows.
- The modular 4-stage architecture utilizes the decision-making capabilities of LLMs effectively, allowing for transparency and localization of error.
- The proposed approach and evaluation includes mitigation strategies, which is key for cybersecurity research.
- The evaluation is expansive, using numerous LLMs to evaluate the proposed approach. The quantitative measures show high performance and stability across various scenarios, demonstrating strong contributions.

**Weaknesses:**

- The core novelty lies in orchestration of the threat prioritization system rather than representation learning and model adaptation. This could be more appropriate for a core cybersecurity research audience. It would be beneficial to incorporate latent representations with adaptor-based finetuning, which could be more useful for the LR audience.
- An ablation study to show what is causing performance gain would be helpful, particularly looking at data enrichment, reasoning and retrieval augmentation individually. Furthermore, a qualitative analysis of LLM attention and reasoning traces would enhance transparency and replicability, and better understanding of the proposed framework.
- Error analysis and failure case analysis would have been useful, particularly to understand robustness to noise (in an adversarial setting), out of distribution generalization etc.

**Questions:**

1. In section 3.3, it is unclear how the “probability of observed exploitation” is computed. Is this from a supervised learning ML model or and LLM?
2. In section 3.4, when migration strategies are produced, is there a difference in performance between long and short sequence of events? Generally, mid-size LLMs have known issues with long sequences.
3. In Figure 3 and 4, it is unclear why specialized LLMs are performing worst than vanilla LLMs. Should it be the other way round, particularly if the specialized LLMs are finetuned for threat detection?
4. - In evaluating the approach, all base-line mechanisms use LLM approaches, though the paper mentions previous approaches that target isolated tasks. Is there a reason why those other non-LLM baseline mechanisms was not used for comparisons?

---

### Official Review · Reviewer_Xpur · 2025-10-30

**Soundness:** 2
**Presentation:** 3
**Contribution:** 2
**Rating:** 4
**Confidence:** 5

**Summary:**

This paper proposes POLAR, a four-stage LLM-based framework designed to automate cyber-threat prioritization. POLAR’s pipeline includes:
* CTI Triage: disentangles raw incidents into structured threat instances and enriches them with metadata (CVE, ATT&CK mappings, KEV status).
* Static Analysis: translates enriched contexts into CVSS metrics using LLM-guided evidence reasoning.
* Exploitation Analysis: predicts exploitation likelihood over the next 30–90 days by reasoning over temporal narratives of disclosure, PoC, and KEV events.

Mitigation Recommendation: retrieves relevant patches and mitigations and ranks them by risk and feasibility.
Experiments use vendor advisories, CISA KEV, Exploit-DB, VirusTotal, and EPSS datasets. POLAR consistently improves F1 and RMSE across triage, static scoring, exploitation forecasting, and mitigation recommendation when coupled with both general-purpose and security-specialized LLMs. The authors release code anonymously.

**Strengths:**

[+] POLAR unifies triage, scoring, forecasting, and mitigation into a single reasoning pipeline

[+] The paper provides concrete prompt templates, workflows, and evaluation breakdowns per stage (A.1–A.4), facilitating reproducibility.

[+] By combining CVSS semantics with CTI feeds and temporal modeling, POLAR produces interpretable outputs that align with analyst practices rather than opaque numeric scores

**Weaknesses:**

[-] The novelty lies more in integration than algorithmic advancement. Although the pipeline is well engineered, many components reuse established paradigms such as RAG, reasoning prompts, and structured extraction.

[-] Insufficient baselines for each sub-task. For example, the exploitation-forecasting results lack comparison with traditional EPSS models. Static analysis is simple NL analysis and is only benchmarked against NVD-assigned scores rather than ML-based severity classifiers.

[-] Scalability and cost not addressed. Each stage involves multiple LLM calls (triage, CVSS metric inference, exploitation narrative reasoning, mitigation retrieval). The paper omits latency, token cost, or throughput analysis

[-] Missing ablation for pipeline coupling. It remains unclear how much each module contributes to the final prioritization accuracy

[-] Missing comparative baselines with related cyber threat prioritization approaches. The authors integrate several exiting techniques, but did not present a comprehensive comparison with these techniques and other related approaches.

**Questions:**

Q1: Exploitation forecasting formulation.
The paper treats prediction as reasoning over temporal narratives. How is supervision provided? Are EPSS probabilities directly used as ground truth, or does POLAR self-generate pseudo-labels from KEV timelines? How does it handle missing or conflicting timestamps?

Q2: Temporal reasoning mechanism.
Does POLAR explicitly encode temporal order (e.g., via JSON timeline -> LLM prompt) or rely purely on narrative reasoning? Have the authors compared with a numeric regression baseline trained on event embeddings?

Q3: CVSS metric inference accuracy.
Appendix A.2 lists eight metric workflows. Were those metrics predicted independently and combined via the official formula, or was there a joint reasoning step ensuring consistency?

Q4: Mitigation ranking reproducibility.
Since prioritization depends on qualitative reasoning over (sₖ, pₖ), how deterministic are outputs across runs? Is there variance analysis or seed control to assess reproducibility?

Q5: Data freshness and bias.
Most evaluation datasets (KEV 2024, Exploit-DB 2025) are public but fast-evolving. Did the authors freeze a snapshot? How would POLAR perform on emerging CVEs unseen during data collection?

Q6: Compare with prior systems.
How does POLAR compare to related work on exploitability prediction and cyber threat prioritization?

---

### Official Review · Reviewer_AKKW · 2025-10-30

**Soundness:** 3
**Presentation:** 3
**Contribution:** 2
**Rating:** 4
**Confidence:** 3

**Summary:**

This paper introduces POLAR, a modular framework leveraging large language models (LLMs) to automate cyber threat intelligence (CTI) analysis and prioritization. POLAR decomposes the problem into four sequential tasks: Triage, Static Analysis, Exploitation Analysis, and Mitigation Recommendation, each tailored with domain-specific prompting and LLM configuration.

**Strengths:**

1.	Automating CTI prioritization is a critical real-world problem with high practical impact. The paper demonstrates how LLMs can effectively support security operations.
2.	The paper conducts an evaluation on real-world threat incidents with high-quality and detailed statistics.

**Weaknesses:**

1.	The primary contributions are in system design and application of existing LLMs to cybersecurity. There is limited novelty in terms of machine learning and AI. The paper may not be well aligned with the scope of ICLR.
2.	The system assumes clean and truthful input reports. In practice, attackers may obfuscate language or inject misleading information to evade detection or reduce threat scores. The paper should provide a discussion or mitigation for such adversarial inputs.
3.	The system performance heavily depends on prompt design and task-specific tuning, but the design rationale for prompts and few-shot examples is not deeply analyzed or generalized.

**Questions:**

1.	What’s the technical novelty of POLAR in terms of machine learning and AI?
2.	How would POLAR handle adversarially crafted incident reports that intentionally obfuscate or mislead? Can LLMs be made robust to such scenarios?

---

### Note · Authors · 2025-12-03

I have read and agree with the venue's withdrawal policy on behalf of myself and my co-authors.